

# Assessing public outreach strategies in cities coping with climate risks

Rosa Vicari[1], Ioulia Tchiguirinskaia[1], Daniel Schertzer[1]

[1]Laboratory HM&Co, U. Paris Est / École des Ponts ParisTech, Champs-sur-Marne, 77455, France

*Correspondence to*: Rosa Vicari (rosa.vicari@enpc.fr)

**Abstract.** The resilience of our cities to weather extremes relies both on physical–environmental factors, as well as socio–economic factors. The last include communication processes among the members of an urban community. This paper presents a study that aims at understanding how to objectively assess the progress that is made through public outreach campaigns on urban resilience. According to this research, seizing the added value of science outreach efforts calls for an

assessment method that takes into consideration the interactions between communication processes and other urban resilience drivers. The paper starts from presenting examples of methods to assess urban resilience to weather extremes. On the basis of this review, some metrics addressing communication issues are selected and first insights are given on evaluation techniques that can be used to understand if and how public outreach activities enhance urban resilience to weather extremes. The paper also outlines examples of target audiences in communication strategies aimed at enhancing resilience. Indeed

effective monitoring of communication impact relies on an accurate audience definition. The paper finally presents a range of experiments led by the HM&Co laboratory to assess communication activities addressed to non–specialised audiences and tailored for an urban flood resilience project. Different assessment methods have been tested to apprehend their strengths and weaknesses in the framework of urban resilience strategies.

**1 Introduction**

The early engineering interpretation of resilience used to be rather concerned by the capacity to absorb a stress and to continue to maintain the function of a system that is assumed to be stable. The social–ecological resilience approach outlined by Holling (1973) and supported by the mathematical model of Ludwig et al. (1997) – departed from the

mainstream interpretation of resilience by pointing at renewal, re–organisation, innovation, development and adaptation after disturbance as important capacities in a resilient system (Gunderson and Holling, 2002; Berkes et al., 2003; Adger, 2006). In this paper we will thoroughly outline the social–ecological resilience approach for two main reasons:

- It emphasizes the human–in–nature perspective, by considering interactions and interdependence among social, economic, physical and environmental components of a social–ecological system;



- It provides definitions of resilience that are a basis to develop operational resilience metrics.

According to this approach, *resilience* can be defined as the capacity of a system to absorb disturbance by continually changing, adapting and reorganizing, and yet to preserve the same function, structure and feedbacks, i.e. maintain its identity

(Folke et al., 2010).

In the social−ecological resilience approach the accent is on thresholds, uncertainty, non−linear dynamics, interplay between gradual change and rapid change  (Walker and Meyers, 2004) and on interactions of these dynamics throughout spatial and temporal scales. Because of the complexity of systems, that continuously evolve and are characterized by multiple interactions across spatial and temporal scales, the trajectory followed by a system after a perturbation cannot be described

with the help of the concept of *stable states* or *equilibriums,* but rather with the concepts of *regimes* or *attractors* (Carpenter, 2003). However, these attractors are more complex than classical attractors, because systems are not only complex in time, but in time and space. Space complexity implies qualitative differences with respect to classical chaos concepts, in particular their intrinsic predictability limits (Schertzer and Lovejoy, 2004). This is presumably the source of the important gaps between theories and applied metrics of resilience. Indeed, operational resilience metrics are usually defined with the help of

semi−quantitative indicators that are applied to variables aggregated up to the outer scale of the system, not across the various spatial scales of the system. It is worth to mention that Tchiguirinskaia et al. (2014) showed that multifractals can be used to defined both resilience and its metrics across space−time scales.

## 2 Implementing social−ecological resilience

By the 2000s increasing attention among academics, as well as practitioners, has been devoted to the implementation of resilience. Putting the concept of social−ecological resilience into practice involves relevant changes in policy and decision− making.  Indeed, the social−ecological resilience approach emphasizes the need to apply the principle of subsidiarity, i.e. to decentralize risk management, to encourage citizen participation and share responsibilities (Tanguy, 2015). Another relevant

aspect specifically concerns urban resilience to climate change:  the social–ecological resilience approach puts the focus on adaptation and mitigation to climate change, rather than on the need to reduce greenhouse gases emissions and urbanization.

Going beyond theory and implementing resilience requires resilience metrics: such indexes allow decision makers to compare the costs of resilience enhancement actions with the economic, environmental, social, and sanitary costs of non−

action. Resilience metrics also help to set up clear objectives at the beginning of a project, to evaluate and improve management capacities, to increase transparency and stakeholder involvement during and after a project. According to Carpenter et al. (2001) quantitative definitions and resilience metrics allow testing hypothesis on the dynamics of systems.





Operational and measurable definitions of resilience should be consistent with the theoretical definitions, and the same definitions, or at least similar definitions, should be applicable to different systems and enable cross−systems comparisons.

Another important step to assess the resilience of systems is to identify the disturbance and the system we are interested in.
This allows defining the spatial, social and temporal scales of the system. However, the question "resilience of what, to what?" (Carpenter, et al. 2001) shouldn't lead to focus on *specified resilience* only, but also to consider *general resilience* that takes into account all parts of a system and all kinds of shocks, also new ones.
Specified resilience entails the risk of enhancing resilience of specific components of a system to specific shocks, while weakening resilience in other ways (Cifdaloz et al., 2010). The *HOT* (*Highly Optimised Tolerance*) theory (Carson and
Doyle, 2000) shows that systems that become highly resistant to frequent types of stresses become vulnerable to infrequent shocks. Hence fostering specified resilience doesn't necessarily avoid regime shifts, but highlights the importance of considering the transformability of a system.

The resilience assessment techniques presented in this paper are extremely heterogeneous in terms of the concept of
resilience they refer to, the system and disturbances they consider, the selection of indicators and variables, the degree of on−site implementation. Examples of resilience indicators presented here include five cases of urban resilience indicators, and three other cases concerning indicators aimed to assess flood resilience in urban areas.

### 3 Examples of methods and metrics that are suitable to assess urban resilience to weather extremes

Among all the methods and metrics presented in the report "Review of alternative approaches to assess resilience to extreme weather" (Vicari et al., 2015), the *Hyogo Framework for Action* is the most widespread assessment approach worldwide, with about 270 municipalities that implemented it (www.preventionweb.net). "Indicators of Progress: Guidance on Measuring the Reduction of Disaster Risks and the Implementation of the Hyogo Framework for Action" was published in 2008 by the ISDR Secretariat (UN/ISDR, 2008), following the request of national governments for a tool to assess their
progress toward the goals of the "Hyogo Framework for Action 2005−2015: Building the Resilience of Nations and Communities to Disasters".
The HFA strategy has a high international relevance and institutional legitimacy. Even though qualitative assessment is widely used, quantitative variables are also considered. An interesting point is the importance given to stakeholder active involvement in the assessment process. However, it should be noted that there are discrepancies between the HFA approach
and the social−ecological resilience perspective: the HFA points at disaster reduction in the context of sustainable development rather than at resilience as an overall objective; furthermore policy making factors prevail on other resilience drivers.




The *MAES – Mapping and Assessment of Ecosystems and their Services* framework – outlined by the European Union (European Commission, 2018) – goes quite the opposite way with a list of indicators for urban ecosystems that pays particular attention to the physical and environmental dimensions of a city. Indeed this document springs from the Target 2 of the EU "Biodiversity Strategy to 2020" (European Commission, 2011) that is to maintain and enhance ecosystem services

in Europe. According to the MAES framework, ecosystems services rely on high ecosystem quantity and quality that are necessary to ensure their resilience.

Nevertheless, the relation between ecosystems and the socio–economic systems is not disregarded: according to the Biodiversity Strategy, ecosystem services increase well–being and have an economic value. The MAES framework specifies that in urban areas good living conditions for humans contribute to establish if urban ecosystems are in good condition. The

relation between ecosystems and socio–economic systems arises also from the fact that human activities are considered as the drivers of change. Other relevant aspects of the MAES framework are that all the indicators are quantifiable, they are scalable from a local to a global scale and allow detecting change over time. Experts with particular knowledge in urban areas were involved to define indicators. EU Member States, other scientific experts and the EC environmental policy units were involved to ensure that indicators are policy relevant.

Stakeholder active participation to the evaluation process is a key aspect of three other assessment methods: the *Integrated Analysis of Territorial Resilience*, the *SMARTeST Indicators of Success*, and the *Four R's – Five C's* approach. The last two were specially designed for flood resilience.

The *Integrated Analysis of Territorial Resilience (AIRT)* was developed by the French Ministry of Ecology, Sustainable Development and Energy (MEDDE) and it led to the analysis of twelve sites affected by disasters to provide recommendations to local stakeholders on how to enhance resilience, adaptation and recovery of the territory.

With a perspective that is in line with the social–ecological resilience approach, the MEDDE emphasizes *resilience* as a key goal. Indeed, "developing sustainable and resilient territories" is one of the axes of the *French strategy of ecological*

*transition for sustainable development (SNTEDD)* adopted by the French government in February 2015 (Tanguy, 2015).

The project entailed a first phase of data collection in the pilot sites through over 60 interviews with the stakeholders. During the second phase of the project three working groups met to discuss about: 1) *The citizen at the heart of resilience*, 2) *The territory of resilience*, 3) *Enhancing resilience: the integration factor in public policies*.

The *SMARTeST Indicators of Success* were defined as a key step of the *Implementation Strategies for Flood Resilience (FRe)* (Tourbier, 2011) management, in the framework of *Smarter Resilience, Tools, Technologies and Systems* (SMARTeST), a European FP7 research project aimed to develop, test and integrate new small–scale solutions to enhance urban flood resilience. According to this approach, the degree of flood resilience in a city can be measured through four sub–indexes corresponding to four different dimensions of an urban system (spatial planning, structural planning, social planning



and risk management). A peculiarity is that the weight of each sub‒index is not pre‒defined but it is considered a critical aspect to be discussed with the stakeholders.

The *Four R's ‒ Five C's* approach was designed to measure flood resilience at community level. The method is outlined in
the white paper "Operationalizing Resilience Against Natural Disaster Risk" (Keating et al., 2014) – one of the key documents of the multi‒year research program *Zurich Flood Resilience Alliance* led by the Zurich Insurance Group. A community is here described as a system with multiple interacting dimensions: the *five community capitals* or *5 C's* that correspond to the *Human*, *Social*, *Physical*, *Natural*, and *Financial capitals*. The resilience of the city relies on four key characteristics that are called the *4 R's* (*Robustness, Redundancy, Resourcefulness, Rapidity*). A notable aspect of this
approach is the use of an interactive web tool in the pilot sites that is aimed to involve community members in the assessment campaigns.

The concept of a system with multiple interacting components is also employed to assess city resilience by *100 Resilient Cities*, a cities network founded by the Rockfeller Foundation, that aims at supporting cities in coping with natural, social
and economical shocks and stresses and that employs the *City Resilience Index* (www.100resilientcities.org; The Rockefeller Foundation and ARUP, 2015; 2017). Similarly to the *Four R's ‒ Five C's* approach, also *100 Resilient Cities* associates urban resilience to other key characteristics: reflective, resourceful, inclusive, integrated, robust, redundant, flexible.

The use of quantitative variables in resilience assessment presents several advantages: numeric data allow reducing and
restructuring a complex problem to a limited number of variables, they can be used for statistical analysis and consequently to generalize a finding, they facilitate inter‒comparison, and allow automated data collection and analysis. Examples of quantitative variables are presented in four cases of resilience assessment methods: the above mentioned *Hyogo Framework for Action* – where quantitative variables are combined with qualitative evaluation – and *MAES framework*, the *Resilience Alliance* approach, the *Baseline Resilience Indicators for Communities* (BRIC), and the *Performance indicators to assess*
*urban networks resilience*.

*Resilience Alliance (RA)* (Resilience Alliance, 2010) is an international, multidisciplinary research organization that develops guidelines and principles to assess resilience of social‒ecological systems and to implement sustainable development strategies. RA outlines an assessment framework that is totally consistent with the social‒ecological approach,
so that the RA evaluation method entails specific and general resilience, as well multiple spatial and temporal interacting scales. According to this method, for each quantitative variable a threshold should be identified as well as the effects if the threshold is crossed.



The *Disaster Resilience Of Place (DROP)* model and the *Baseline Resilience Indicators for Communities (BRIC)* are defined by Cutter (Cutter et al. 2008; 2010) who focuses on resilience to natural hazards at community level and on the relationship between resilience and vulnerability. The model is a conceptual basis to identify resilience indicators that can be used with different spatial scales. Like in the SMARTeST proposal, the *DROP* and *BRIC* approach defines a composite resilience

index, with sub‒indexes corresponding to different dimensions of the urban system. However, unlike SMARTeST, Cutter exclusively considers quantitative variables, and he defines a method to normalize different ranges of values to a unique scale.

Another method to normalize different ranges of values corresponding to different quantitative variables is described by

Serge Lhomme (Lhomme et al., 2013) who defines *Performance indicators to assess urban networks resilience*. Lhomme aggregates three sub‒indexes, corresponding to three different resilience capacities: absorption, resistance and recovery.

### 4 Assessing communication for resilient cities to weather extremes

The key role played by social networks and information flows is a recurring topic that is common to all the assessment methods presented in this paper. From a complex system perspective, communication among stakeholders affects resilience, since it impacts on the system dynamics, its adaptability and transformability[1].

According to this study resilience indicators shouldn't only consider communication infrastructures but should also assess

communication processes and their interactions with other resilience drivers. As it is stated by Charrière et al. (2017), impact assessment of risk communication campaigns isn't a widespread practice yet. Nevertheless, some of the existing resilience assessment approaches consider communication as a key factor. For instance, in the HFA methodology one of the five priorities for action is to "use knowledge, innovation and education to build a culture of safety and resilience at all levels" (UN/ISDR, 2010). In other examples of evaluation methods communication indicators are considered but remain marginal:

this is the case of the *BRIC framework* ‒ where the percentage of population with a telephone is used as an indicator of

---

[1] *Adaptability* is a characteristic of systems that is considered by Folke (Folke et al., 2010) as part of resilience. Folke defines adaptability as "the capacity to adjust responses to changing external drivers and internal processes and thereby allow for development along the current trajectory (stability domain)". According to Folke, adaptability of systems should be addressed to understand their dynamics and development, as well as transformability, another characteristic that is interrelated to resilience. *Transformability* is defined as "the capacity to cross thresholds into new development trajectories" and Folke highlights that transformation is necessary to resilience since "transformational change at smaller scales enables resilience at larger scales".



communication capacity (Cutter et al., 2010) – or of the *RA assessment method* that applies social networks mapping as part of the governance system analysis (Resilience Alliance, 2010).

Public outreach impact can be evaluated in terms of quantity and quality. Quantity can be considered as corresponding to
the communication frequency and audience size. For instance we performed a four years monitoring of the communication frequency and audience size in the framework of the Interreg IVB RainGain project. As it was required by the Interreg NWE IVB funding program we systematically reported the number of website unique visitors, events and participants, publications and readerships, press articles and readerships, TV/radio reports and audience size. These data present the advantage that collecting them requires minor logistical efforts, and can be done after the communication activity is over. This type of
evaluation provides an insight on how communication efforts and resources are distributed and on the resonance of each communication activity. Furthermore, it facilitates comparison of data over time and across different experiences. However, as it is analysed thoroughly in Sect. 6, frequency and audience size monitoring was not sufficient to appreciate how far communication contributed to achieve the main project goal of enhancing urban resilience to floods: assessing communication quality was essential.

Evaluating the quality of communication raises the question on how to define and identify a positive or negative performance. The project goals and the target audiences, when they are clearly defined, can be the guiding criteria. In other words, the communication activity outcome can be considered as positive if it has contributed to reach the target audience and achieve the project goals. In the RainGain project we opted for an evaluation based on a survey (see Sect. 6), however various qualitative and quantitative analysis methods exist. Open‒ended questionnaires, unstructured interviews,
observations, and focus groups are qualitative methods that allow descriptive data collection. Surveys allow collecting data in numerical form that can be categorized and analysed using statistics. Text mining applied to web contents can also provide quantifiable information. As it is claimed by Topping and Illingworth (2016) social media generate a large amount of data that allow training computer aided analysis tools and obtain accurate and consistent assessments of users opinions. Some examples of quantitative analysis of Web communication trends through advanced text mining tools are:

-    The *Europe Media Monitor* (emm.newsbrief.eu) provides advanced analysis, generated by software algorithms, for monitoring of both traditional and social media;

-    The *Science in the Media Monitor* (www.observa.it/science-in-the-media-monitor/?lang=en) provides a computer aided analysis of coverage of topics related to scientific research and technological innovation in the Italian online newspapers and blogs;

-    In the framework of the Bologna City Branding Project, sentiment analysis was applied to social media to investigate how different target audiences perceive the city of Bologna (R. Grandi and F. Neri, 2014).

In the field of hydrology and meteorology, and more in general in the science and technology sphere, access to information has hugely increased in terms of variety and quantity, as a consequence of different factors, among others the development of public relations by research institutes and the pervasive role of digital media (Bucchi, 2013; Trench, 2008).



Thanks to the exploration techniques of unstructured Big Data – such as the above mentioned tools – it is possible to navigate through information databases and to study the trajectories formed by unthinkable amounts of written and oral texts, images, audio‑visual contents, links that are produced and spread through websites, blogs, social networks, press releases, press articles, publications, etc.

Digital innovation makes possible a "navigational practice" through datasets "without making the distinction between the level of individual component and that of aggregated structure. It becomes possible to give some credibility to Tarde's strange notion of monads", intended as "a point of view on all the other entities taken severally and not as a totality" (Latour, 2012). These analysis techniques have been explored not only in sociology, but also in mathematics, in particular by Peter Grindrod (Grindrod, 2011).

## 5 The target audience in resilience communication

Any communication strategy, as well as its assessment, requires to define the target audience and to profile it. Defining the target audiences – i.e. groups to which communications are addressed – is a crucial phase of a communication plan that will

influence the strategy efficiency, especially when selecting messages, communication means, time and locations.

Examples of communication strategies aimed to enhance resilience and the corresponding target audiences are provided by Reef Resilience (www.reefresilience.org), the London Resilience Partnership (Ingleby, 2014), CEPRI (European Centre on Risk Prevention) (CEPRI, 2011), TOMACS (Tokyo Metropolitan Area Convection Study for Extreme Weather Resilient

Cities) (Tsuyoshi et al., 2015), CASA (Collaborative Adaptive Sensing of the Atmosphere) (Chandrasekar et al., 2012; Chandrasekar and the full DFW team, 2015; Donner et al. 2012), Wikiresilience (wikiresilience.developpement-durable.gouv.fr), and the INTERREG NWE IVB RainGain project (Vicari et al., 2015).

As it is highlighted in Table 1, various criteria can contribute to the target audience definition.

As a generally valid principle, selection criteria should be based on the communication strategy objectives that in turn

depend on the resilience management objectives. A detailed definition and profiling of the audience groups is another important point, Quite often communication strategies refer to the *general public* as a target, while the more an audience profiling is accurate the better communication activities can be tailored and directed where a real need exists and results can be detected. Furthermore, it will facilitate monitoring of progress towards the communication goals.




|  | Audience definition criteria | Target audiences |
|---|---|---|
| **Reef Resilience** | - Risk causes<br>- Interest for resilience enhancement<br>- Positive or negative influence on the rest of the community<br>- Benefits and disadvantages of resilience enhancement<br>- Uses of natural resources at risk | |
| **London Resilience Partnership** | The impact that an incident has on groups of people | - People directly affected by the emergency<br>- Local people, friends and relatives,<br>- Wider audience |
| **CEPRI guide on risk communication** | Awareness and involvement in flood issues | - Participants<br>- Early adopters<br>- Early majority<br>- Late majority<br>- Obstinate skeptics |
| **CASA** | - Capacity to understand and use the information<br>- The impact that an incident has on groups of people | - Linguistic groups<br>- Minorities<br>- Communities at various levels of risk |
| **TOMACS** | - Capacity to understand and use the information<br>- The impact that an incident has on groups of people | - Fire brigades<br>- Public transport companies<br>- Residents of at-risk urban areas |
| **Wikiresilience** | Degree of awareness and kind of involvement in resilience issues | - Citizens<br>- Local associations<br>- Public authorities<br>- Practitioners from the private or public sector<br>- Researchers, teachers, students, institutes |
| **RainGain** | Degree of awareness and type of operational involvement in urban flood resilience management | - Politicians,<br>- Policy and decision makers for Urban water management at national, regional and local level;<br>- Local and regional government entities;<br>- Water authorities and water utilities; |





|  |  |
| --- | --- |
| - | Weather services; |
| - | General public; |
| - | Partners of other projects. |

Table 1: Comparison of different approaches to target profiling in resilience communication activities.

**6 The RainGain project: experiences and perspectives in communication assessment for urban resilience projects**

The Hydrology Meteorology and Complexity (HM&Co) laboratory coordinates several research projects aimed to enhance urban resilience to weather extremes. HM&Co research projects also involve developing and strengthening a network of stakeholders through dissemination and public engagement activities.  Since 2012, HM&Co has made efforts in this direction. It was first involved in the participatory workshops addressed to the stakeholders of the FP7 SMARTeST project. The purpose of the workshops was to gather inputs to improve a decision support tool for flood management. After this first

outreach experience, HM&Co coordinated a four years long communication strategy in the framework of the Interreg NWE IVB RainGain project. The main communication objective was "to disseminate and make available the tools and methodologies developed in the project, so that its target groups are informed, educated, involved and mobilized so that vulnerability to urban pluvial flooding is reduced and resilience is enhanced" (Interreg NWE IVB RainGain, 2011). This general goal was further detailed in internal communication goals and external communication goals.

The frequency of communication activities and their impact, in terms of audience size, were monitored since the beginning of the communication plan. This enabled HM&Co to adjust the communication activities during the project implementation if problems were revealed. Indeed, precise target values had been established when the communication plan was designed. During the execution of the plan, the target values were periodically compared with the attained values in order to appraise if

insufficient efforts and resources had been devoted to specific activities (Fig. 1). An interesting output of the communication monitoring reports concerns the press coverage of the RainGain project.





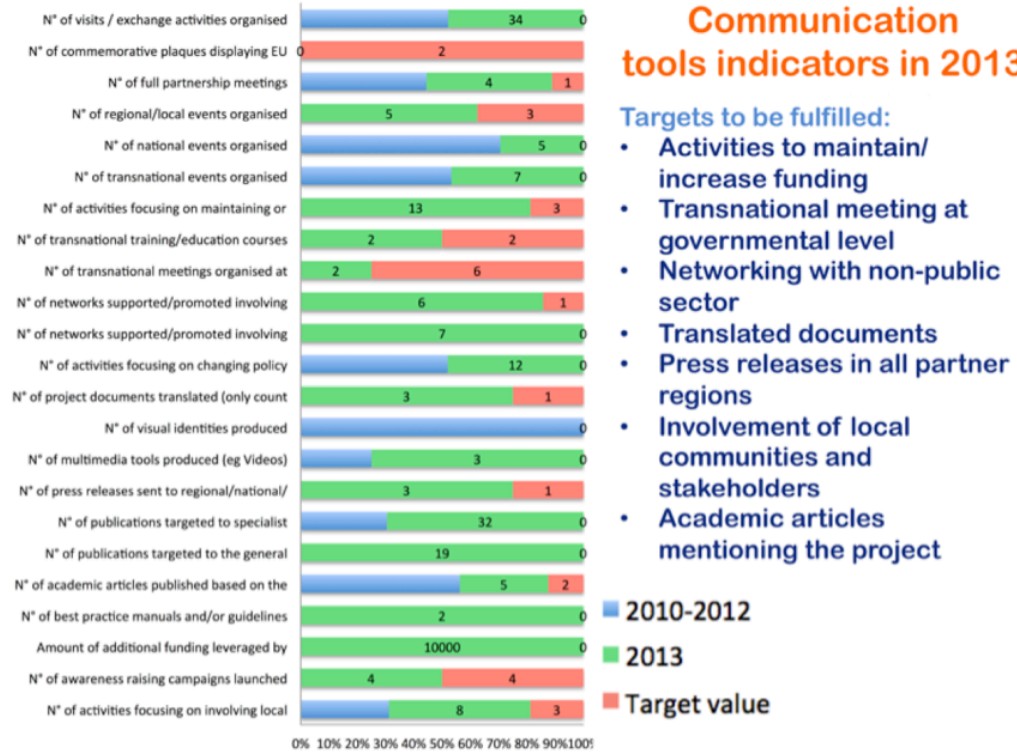

**Figure 1: During the execution of the RainGain communication plan, the target values were periodically compared with the attained value.**

5  Figure 2 shows that from July 2011 to November 2015 at least two news per month – concerning the RainGain project – have been published on printed press, online press, or broadcasted on TV and radio. It can be observed that during specific months, in the project period, the number of news was above two. Two kinds of events occurred when the rate of news was above two per month:

- Communication activities (RainGain press releases and conferences) that can be considered as social and
10      endogenous causes of news rate increase, since they are the output of the project coordinators and communication officer work;



- Flood events in North−Western Europe (in particular the flood that hit South−East of France in October 2015) that we can define as environmental and exogenous causes. The impact of flood events on media coverage is an example of possible correlation between an environmental factor and a social factor that could be further investigated.

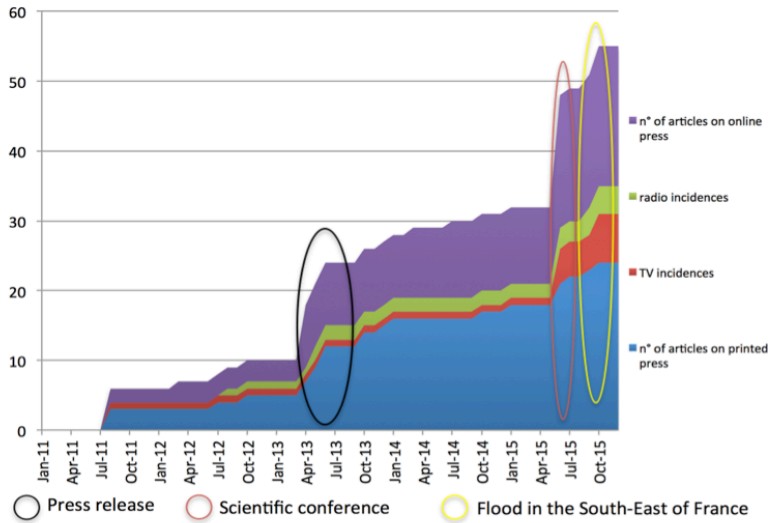

**Figure 2: Number of news (printed press, online press, TV and radio) concerning the RainGain project published from July 2011 to November. The number of news rapidly raised during three specific events: 1) dissemination of a press release on the project; 2) organisation of a scientific conference related to the project ; 3) a flood event in South−Eastern France.**

Data on the number of printed press news have been compared to the data on media audience size. As Figure 3 shows, the ratio between number of news and the audience size is variable: indeed, the audience of local press is limited if compared with the national press; similarly, specialized press has smaller readership than non−specialized press. This variability obviously concerns the printed press, as well as TV, radio and online press.

Another aspect that inevitably impacts news visibility, and should be taken into account, is the size (or duration in the case of TV and radio) and the position (or timeslot) of news.





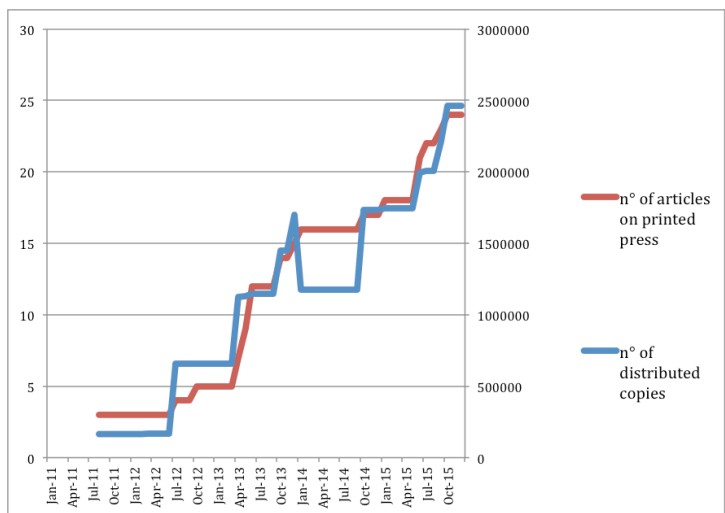

**Figure 3: The ratio between number of news and the audience size: the variability is due to the fact that the audience of local press is limited if compared with the national press; similarly, specialized press has smaller readership than non−specialized press.**

Data on communication frequency and audience size do not capture the quality of the communication contents. For instance, the above data do not show that the RainGain communication activities allowed putting under the spotlight not only flood risks, but also the opportunities offered by research and innovation for smart cities development. Bad news often dominates press headlines, but the news selected in this research can be considered an exception to the rule.

Besides appraising the frequency and the contents of a message, it is also relevant to analyse if it is understood and accepted by the audience. A first attempt to answer to these questions has been made in the framework of an exhibition dedicated to the RainGain project that was held in April 2014. After the event, a survey was distributed to the visitors with the aim of exploring if the exhibition was understood and if it changed the visitor perception of the RainGain project (Persoz, 2014).
Figures 4(a), 4(b) and 4(c) present the answers to three of the survey questions aimed to test the knowledge of the visitors
after the exhibition. For this first evaluation a small sample was selected, since the main aim was to test a methodology that can be applied to different case studies. In order to perform a comparative experiment, a control group of respondents, who didn't attend the exhibition, was selected. The control group included respondents who already heard about the project and





respondents who never heard about it. Some experts were among the respondents but they have been excluded from the sample in order to obtain a relative homogeneity in terms of background knowledge.

**(a)**

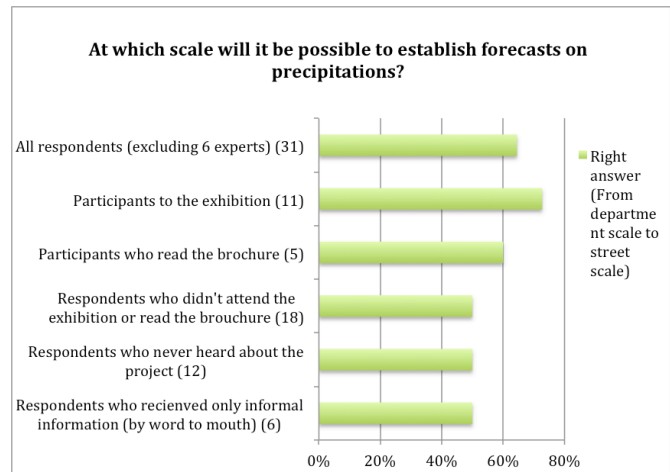

**(b)**

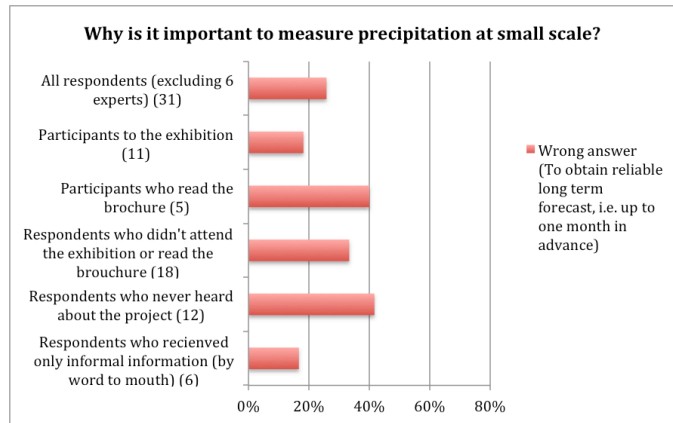





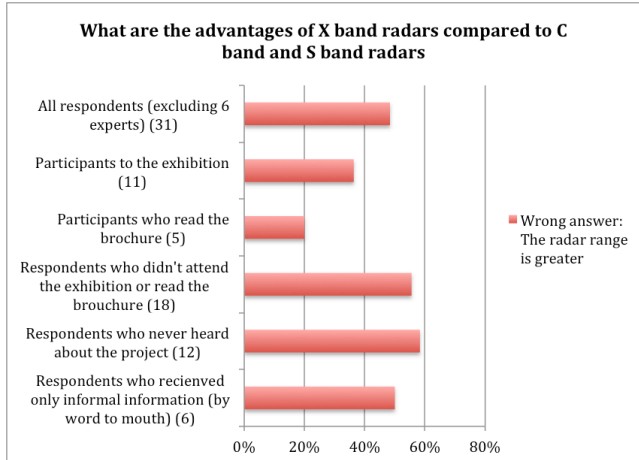

**(c)**

**Figure 4: The answers to three of the survey questions aimed to test the knowledge of the visitors after the RainGain exhibition held in April 2014.**

Figure 4(a) shows that the number of exhibition visitors ticking the correct option for the question "At which scale will it be possible to establish forecasts on precipitations?" is 23% higher than in the control group. As it appears in Figure 4(b), the number of visitors providing wrong responses to the question "Why is it important to measure precipitations at small scale?" is 15% lower than in the control group. According to the results presented in Figure 4(c), the wrong responses to the question

10 "What are the advantages of X band radars compared to C band and S band radars" are 20 % less frequent among the exhibition visitors. The discrepancy between the visitors' results and the control group results is between 15 % and 23 % and it provides an approximate indication of the impact of the exhibition in terms of knowledge dissemination.

An unexpected result concerns the responses of the exhibition visitors who read the brochure in Figure 4(a) and 4(b). In Figure 4(a) the rate of correct responses of the visitors who read the brochure is 60 %, while in the whole group of visitors it

15 is 73% and it rises up to 80% if we consider only those visitors who didn't read the brochure. Figure 4(b) shows that the rate of wrong answers among the visitors who read the brochures is surprisingly high: it is nearly comparable with the rate of wrong answers of the respondents who never heard about the project. A plausible explanation is that the visitors who picked the brochure spent small time to read the exhibition panels and that part of the brochure information was not enough didactic and suitable for the general public.





Figure 4(b) highlights another interesting result: the lowest rate of wrong answers corresponds to the group of respondents who didn't attend the exhibition but heard about the project. According to this figure, face–to–face communication can strongly reinforce transmission of highly technical information.

5    Figure 5 presents the answers to a survey question aimed to evaluate the visitor risk perception after the exhibition and if this event reinforced the project acceptance. The results show that the exhibition and the brochure, i.e. formal and official information, helped to reassure the visitors on security issues. On the contrary, it seems that word of mouth communication tended to rather compromise the achievement of the project goals.

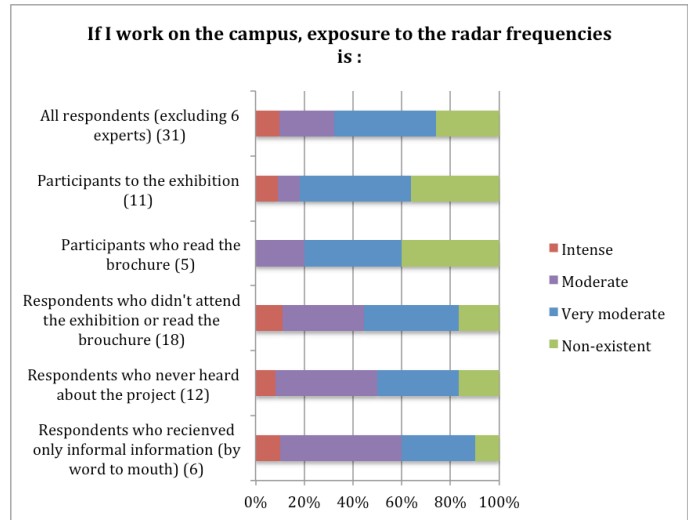

**Figure 5: answers to a survey question evaluating the risk perception of the visitors after the RainGain exhibition.**

While surveys with close–ended questions allow quantifying the results, interviews can reveal more insights on the reactions and reasoning of the respondents. Two assessments based on open–ended questions were conducted during the RainGain

15    project to evaluate the impact of outreach activities.

One of the achievements of the RainGain project was the inauguration of the new high–resolution weather radar at École des Ponts ParisTech in June 2015, followed by an International Conference related to the COP21 (2015 United Nations Climate Change Conference). The promotion of these events involved a wide range of outreach activities and

20    means. One of the promotional contents that were produced on this occasion was a short video (Mulard et al., 2015)





that shows the installation of the radar, highlights the importance of this device in terms of research and innovation and invites the audience to attend the conference. The video was mainly addressed to the students and workers of École des Ponts, since the school building is located in front of the radar site. The manager of the school café, a charismatic and well–known figure in the campus, was involved as the speaker of the video to catch the attention of the audience. While the video was broadcast on Youtube and on some of the school screens, four students were interviewed to have an insight of what kind of information they expected and how they interpreted the video contents.

| |
| --- |
| What was unclear in the video and why? |
| Which aspects of the project would you like to learn more about? |
| What are the strengths and weaknesses of this video? |

**Table 2: Questions to the audience of the video "Jeanine presents the radar" (Mulard, 2015).**

The video was appreciated by the respondents who found it « catchy » thanks to its dynamic pacing and the speaker. They found also interesting the images of the radar installation. However, the respondents, who were all engineering school students, would have expected more information about the radar functioning and its concrete applications. The respondents were curious about the extent of implementation and impact of the project ("Is the radar already operational?", "How many new radars will be installed in Europe?", "You should include a map with the pilot sites in the video"), the radar functioning ("Does the radar allow predicting the rainfall volume?", "It would had been nice see some radar images" ), the  researchers and engineers who will operate it, the services that will be developed with these new weather data ("Is it used only for weather forecast ?", "Is it possible to use them for Roland –Garros ?"). These results will be used to design new surveys that are addressed to students from an engineering school. For instance it appears relevant to include questions that make the link between a flood resilience project and their professional interests and that are accurately tailored to their background knowledge.

A similar assessment based on three open–ended questions (Table 3) was undertaken in November 2015 to evaluate the impact of a workshop on the RainGain Project in the framework of the Provin Climate Forum. A non–traditional method such as snapshot interviews (Fogg Rogers et al., 2015) was chosen to overcome the constraints inherent to evaluating a young audience – a group of 20 eight–year–old school pupils – in the context of a public outreach forum. The assessment highlighted that the audience enjoyed and memorized very well a manual activity on rainfall observation where they





were active participants. The interviews also highlighted that while the purpose of the third question was to assess the clarity and exhaustiveness of the communication contents, the respondents understood it as a question testing their learning capacities. This result suggests that questions addressed to a young audience should be formulated in such a way that the respondents don't' feel like they are being examined.

| |
|---|
| What did you like in this workshop? |
| What did you learn that you didn't know before? |
| Is there anything you didn't understand or you'd like to learn more about? |

**Table 3: Questions to the participants at the RainGain workshop held by Auguste Gires in the framework of the Provin Forum (November 2015).**

### 7 Conclusions

According to this study communication assessment should be included in an alternative approach to evaluate urban resilience to weather extremes. Following the review of resilience studies and existing resilience metrics, this research aims to select a range of resilience indicators that focus on communication processes and their impact on urban resilience to weather extremes.

The press impact assessment (Fig. 2 and 3) provides an example of how press frequency and audience size can express the amount of information that has been spread by the press on flood resilience. This evaluation shows how the physical–environmental system (a flood event) can impact the social–economic system (press communication) and that quantitative analysis can be applied to understand such interactions. It would be significant to investigate the contents that spread through the press news: for instance, if the representation of scientific innovation by the

press is positive or negative and what are the correlations with the concept of *resilient city*. Computer–assisted text mining tools are a possible methodological path to be followed: content analysis can be used to evaluate the frequency of specific topics in the press or on the Web; another interesting approach is sentiment analysis (or opinion mining), a big data exploration technique that is used to monitor positive or negative comments on specific issues. Furthermore, network representation of terms co–occurrences allows a reconstruction of cognitive dynamics

through the Web, in other words how far specific information is accepted by the Web users and who are the main opinion leaders.





The visitor attitude (knowledge acquisition and risk perception) described in the exhibition experiment (Fig. 4, 5) cannot be generalized to all the visitors, since the size of the sample is limited. However, thanks to this experiment it was possible to design and test the implementation of a quantitative approach to assess communication quality, specifically aimed to evaluate to what extent a message has been understood and accepted by non-specialised

audiences. Furthermore, the experiment showed that a comparison between the visitor response and the control group response allows a normalization of the response ratings to different questions. As it was previously mentioned, quantitative methods such as surveys, allow statistical analysis and generalization of the findings, inter-comparison across time and case studies, automated data collection and analysis. However a preliminary qualitative study, for example a stakeholder consultation through interviews, is valuable to select a representative sample of the audience,

develop the content of the questionnaire and ensure that questions are formulated in an appropriate fashion.

The media impact monitoring and a visitors' survey – are two examples of how quantitative analysis can be used to study communication processes. Designing quantitative methods to explore communication processes is a necessary step to identify quantitative variables as communication indicators. Once communication indicators will be identified

it will be then possible to study the interactions with other resilience indexes.
The selected communication indicators will include metrics that are suitable for assessing:
- Existing communication processes and interplay with other resilience drivers;
- The need to manage communication processes to improve resilience versus the costs of non-action;
- The strengths and weaknesses of strategies aimed to improve communication processes;
- The added value of public engagement strategies.

Interesting inputs on public engagement for resilience are provided by those evaluation methods that exploit resilience assessment campaigns as an opportunity to enhance dialogue with and among stakeholders. This is the case of the *Iterative Risk Management (IRM)*, a methodology proposed by Zurich Resilience Flood Alliance to merge

expert risk analysis with stakeholder perspectives in the framework of *4R–5C* assessments. The community is involved in assessing performance and potential risks and in identifying and implementing solutions to enhance the *four R's*. IRM approaches are considered a key tool to cope with risk management problems such as "lack of robust data, long time scales, uncertainty in future conditions, operationalization and quantification" (Keating et al., 2014). The Zurich Flood Resilience Alliance recently implemented mobile data collection and web-based assessment: these

tools allow automated collection and analysis of data and thanks to a user-friendly interface stakeholder involvement can be facilitated.
The urban community can also be actively involved during an extreme weather event to enhance the city resilience. In the framework of the Urban Weather Sensing Lab in Amsterdam, citizens' observations were collected through a



smartphone app and combined with rainfall sensors data. The aim of the project was to use these street level data to improve real–time warning and support flood risk management decisions (Koole et al., 2015).

Another attempt to enhance public engagement in resilience projects is the case of the Wikiresilience platform (wikiresilience.developpement-durable.gouv.fr) where the users share their knowledge on resilience and the related

5  field experiences. The website is very rich in terms of contents but the users have not yet contributed to the discussion areas that are currently empty. The experience of Wikiresilience clearly shows that one of the main challenges of collaborative web platforms is to go beyond information co–production and enhance a real dialogue among stakeholders.

10  **Acknowledgements**

This research was performed in the framework of the Chair "Hydrology for Resilient Cities" supported by Veolia and the Interreg NWE IVB RainGain project.

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

Wikiresilience: wikiresilience.developpement-durable.gouv.fr, last access: 10 May 2018.