# Peer review of "Assessing the impact of outreach strategies in cities coping with climate risks"

_Geoscience Communication, 2018_

## Referee Comment (RC1) · Anonymous Referee #1 · 12 Jun 2018

General Comments: The paper asks some interesting questions, but fails to deliver. The writing is confusing and the arguments don't flow very well.

Good framing of resilience and a fair amount of evidence from the literature.

The methodology used is not clear at all, it's very vague. For example, there is no information on the sample used. For example, p13: "For this first evaluation a small sample was selected, since the main aim was to test a methodology that can be applied to different case studies. In order to perform a comparative experiment, a control group of respondents, who didn't attend the exhibition, was selected. " What is the sample size? How were the respondents identified and recruited? How many in the control group?

[Figure]

Specific Comments: Title - it would be good to clarify what exactly are the authors assessing. Is it the impact of outreach strategies? Something else?

The authors need to tidy-up the introduction a bit and make better use of paragraphs, for example. The text doesn't flow as well as it could if good punctuation, paragraphs, etc. was used.

Figure 1: this needs to be re-done. It's very hard to understand who much each bar accounts for as there's no scale. Also, you can't read the Y in full, for example "number of academic articles published based on" - based on what?? Overall, all figures are confusing and difficult to interpret. The quality of the figures is not appropriate for publishing, they are more report-style.

References are included in the conclusions section, as well as refs to figures - this is not appropriate for an academic journal.

Technical corrections: Overall the paper would benefit from being reviewed by a native English speaker.

---

## Referee Comment (RC2) · Anonymous Referee #2 · 13 Jun 2018

This manuscript set out to assess how science communication can contribute to urban resilience, using empirical studies to demonstrate how communication can increase scientific impact on resilience according to one or more resilience assessment frameworks. This is promising goal, and a potentially important contribution to the scholarship; enriching these frameworks with a more carefully consideration for communication. But, excepting a short discussion in the conclusion, unfortunately the manuscript fails to properly establish this link. It looses its way, its significance, and the reader.

In my opinion this paper could be publishable subject to major revisions around its original purpose. More specifically: Abstract: There are two words here that jar with science communication scholarship. 'Progress' to resilience is sticky: when resilience is a moving target, progress is too. 'Objectively' is similarly sticky. Given communica-

tion is often a two-way communication, the communicator is actively embedded in the communication act, and it is then difficult for them to objectively 'step outside' of this communication. Suffice to assess, without emphasising the objectivity.

The introduction...is not an introduction. It does not introduce the work that will be done in the paper, including a clear statement of the research question or aim. This material can be moved to the background Section 2.

Section 2: interesting, but there is no mention of science communication. The focus is on resilience assessment metrics alone. I missed some up-front reflections on all of the unquantifiable elements of resilience. This kind of quantitative work seems, to me, to draw a very narrow concept of resilience.

Section 3 is unnecessarily long and doesn't contribute much to this papers discussion. It is enough to note that there are many different frameworks available, and which frameworks this manuscript focussed to evaluate the impact of science communication. It could be shortened and combined with Section 2.

Section 4 is interesting for this paper; to me it is the core. It should provide more detail on the mechanisms between communication and resilience, to help us better understand the assessment in terms of influence, rather than as bald measurements that reveal no clear causal link between the communication and resilience. I expected to see how communication assessment linked to the metrics of one or more resilience assessment frameworks, but it was missing. Some of the reflections from the conclusion could be moved up here.

Section 5 does not add much. It should be linked to the manuscripts core work, or removed.

Section 6 fails to make any link to resilience assessment metrics. It reads independent to the rest of the paper, as an account of three different communication assessment approaches for a resilience project. I could see how this communication assessment

contributed to resilience as framed in a resilience assessment framework.

To the three methods, the quantitative work I understood, apart from Figure 3. Of course its limited in what it shows, but the authors admit that. The qualitative assessment is more problematic, as presented now. These kinds of 'science comprehension tests' have been around a long time in the communication scholarship, but there was no reflection here on the multiple problems with this approach. For me, the most interesting approach was the interviews, which provides the space for target audiences to explain what they understood, how it could help them or not, and yes, any contributions to resilience as defined in resilience assessment frameworks. This seems, to me, to be the best method for linking communication assessment to resilience assessment.
* * *

---

## Author Comment (AC1) · 20 Jun 2018

We would like to thank Reviewer 1 for his time to review our manuscript and to provide valuable comments and suggestions. We will address all issues raised in the critique and we believe that our manuscript will be much stronger after addressing these comments. Hopefully the changes implemented will satisfy his requirements. Here we would like to list our preliminary responses to the items raised by the Reviewer:

1)

"The paper asks some interesting questions, but fails to deliver. The writing is confusing and the arguments don't flow very well. Good framing of resilience and a fair amount of evidence from the literature. The methodology used is not clear at all, it's very vague."

[Figure]

We appreciate that the Reviewer expressed his interest for the research questions addressed in this paper. We believe that the paper doesn't fail to deliver, since the objective of this research is:

- To compare different techniques to assess communication quality and quantity;

- And highlight their strengths and weaknesses in the operational context of a communication campaign on flood resilience.

However we recognise that the following additions to the text would improve the clarity of the paper:

- Adding a presentation of the research rationale, purpose and questions in the Introduction;

- Adding a section dedicated to the methodology;

- Adding a section focusing on the results and discussion.

2)

"For example, there is no information on the sample used. For example, p13: "For this first evaluation a small sample was selected, since the main aim was to test a methodology that can be applied to different case studies. In order to perform a comparative experiment, a control group of respondents, who didn't attend the exhibition, was selected. " What is the sample size? How were the respondents identified and recruited? How many in the control group?"

As highlighted by the Reviewer, some of the information on the methodology is not specified or it's only indicated in the figures and not in the main text. The new section on the methodology will include information on the size of the sample and the control group, as well as on the identification and recruitment processes for each experiment presented in the paper.

3)

"Title - it would be good to clarify what exactly are the authors assessing. Is it the impact of outreach strategies? Something else?"

We understand the Reviewer concern with the title, we are considering to replace it with "Assessing the impact of outreach strategies in cities coping with climate risks".

4)

"The authors need to tidy-up the introduction a bit and make better use of paragraphs, for example. The text doesn't flow as well as it could if good punctuation, paragraphs, etc. was used."

We also understand his concern about the text of the Introduction and we intend to review the punctuation, paragraph divisions and those sentences that might be unclear.

5)

"Figure 1: this needs to be re-done. It's very hard to understand who much each bar accounts for as there's no scale. Also, you can't read the Y in full, for example "number of academic articles published based on" - based on what?? Overall, all figures are confusing and difficult to interpret. The quality of the figures is not appropriate for publishing, they are more report-style."

We agree with the Reviewer that Figure 1 should be improved by adding the amounts concerning 2010-2012 (that are currently missing) and by checking that all the text in the chart is readable. Concerning the other figures, we will verify if all the information is complete and readable, and if the legends are sufficiently clear.

6)

"References are included in the conclusions section, as well as refs to figures - this is not appropriate for an academic journal."

We thank the Reviewer for pointing out that references shouldn't be included in the conclusions. We intend to overcome this problem by dividing the text of the conclusions
in two parts:

- A section dedicated to the Results and Discussion (that will include the references);

- A section dedicated to the Perspectives (without references).

7)

"Technical corrections: Overall the paper would benefit from being reviewed by a native English speaker"

We are in absolute agreement with the Reviewer concerning the need to benefit from a copy-editing service since the authors aren't English native speakers.

---

## Author Comment (AC2) · 20 Jun 2018

The authors would like to thank Referee 2 for reviewing the manuscript and for providing the authors with his constructive remarks and recommendations. We have considered all issues raised in his critique and we believe that our manuscript will be improved after addressing these valuable comments. Here is a list of our preliminary responses to his comments and we hope that these changes will satisfy his requirements:

1)

"This manuscript set out to assess how science communication can contribute to urban resilience, using empirical studies to demonstrate how communication can increase scientific impact on resilience according to one or more resilience assessment frame-

works. This is promising goal, and a potentially important contribution to the scholarship; enriching these frameworks with a more carefully consideration for communication. But, excepting a short discussion in the conclusion, unfortunately the manuscript fails to properly establish this link. It looses its way, its significance, and the reader. In my opinion this paper could be publishable subject to major revisions around its original purpose."

We are happy to see the Reviewer interest for the purpose of this research. We think that the paper establishes a link between resilient metrics and communication impact assessment, but it could be made more explicit in the Introduction, in Section 4 and in the Conclusions.

2)

"There are two words here that jar with science communication scholarship. 'Progress' to resilience is sticky: when resilience is a moving target, progress is too. 'Objectively' is similarly sticky. Given communication is often a two-way communication, the communicator is actively embedded in the communication act, and it is then difficult for them to objectively 'step outside' of this communication. Suffice to assess, without emphasising the objectivity."

As highlighted by the Reviewer, the word "objectively" could be removed from the Abstract as it is controversial. We also understand his concern with the word "progress", but in our view it is necessary and consistent with the intent of moving from a theoretical to a practical approach to social-ecological resilience. In our view adaptability and transformability are two essential characteristics of a resilient system. However, once it is clear what is the system and what is the disturbance we are interested in, it is possible to establish resilience metrics and compare the corresponding variables in different systems (e.g. different cities) or in the same system in two different moments.

3)

"The introduction...is not an introduction. It does not introduce the work that will be done in the paper, including a clear statement of the research question or aim. This material can be moved to the background Section 2."

We agree with the Reviewer that the Introduction would benefit of a presentation of the rationale, the purpose of this research, and the research questions. As suggested by the Reviewer, some information that is currently in the Introduction can be moved to Section 2.

4)

"Section 2: interesting, but there is no mention of science communication. The focus is on resilience assessment metrics alone. I missed some up-front reflections on all of the unquantifiable elements of resilience. This kind of quantitative work seems, to me, to draw a very narrow concept of resilience. Section 3 is unnecessarily long and doesn't contribute much to this papers discussion. It is enough to note that there are many different frameworks available, and which frameworks this manuscript focussed to evaluate the impact of science communication. It could be shortened and combined with Section 2."

We also agree with the reviewer that sections 2 and 3 could be improved by making more explicit the link with science communication. However, in our opinion, it is important to keep the focus on quantitative assessment, since it facilitates an analysis of the interactions between the socio-economic factors, such as communication, and the physical-environmental factors. In our view, qualitative assessments give valuable insights on resilience but, for the purposes of this research, we prefer to employ it to validate the results based on quantitative assessments.

5)

"Section 4 is interesting for this paper; to me it is the core. It should provide more detail on the mechanisms between communication and resilience, to help us better understand the assessment in terms of influence, rather than as bald measurements that reveal no clear causal link between the communication and resilience. I expected to see how communication assessment linked to the metrics of one or more resilience assessment frameworks, but it was missing. Some of the reflections from the conclusion could be moved up here."

We understand the Reviewer concern with Section 4: we plan to strengthen the links with the other sections and develop the reflection on how communication impacts on resilience with concrete examples from Paris Region. We will also include a paragraph with a list of guiding criteria to assess the impact of communication on resilience. This list was outlined on the base of the review of resilience assessment frameworks in Section 3.

6)

"Section 5 does not add much. It should be linked to the manuscripts core work, or removed."

We agree with the Reviewer that the link between Section 5 and the core of the manuscript should be made more explicit.

7)

"Section 6 fails to make any link to resilience assessment metrics. It reads independent to the rest of the paper, as an account of three different communication assessment approaches for a resilience project. I could see how this communication assessment contributed to resilience as framed in a resilience assessment framework. To the three methods, the quantitative work I understood, apart from Figure 3. Of course its limited in what it shows, but the authors admit that. The qualitative assessment is more problematic, as presented now. These kinds of 'science comprehension tests' have been around a long time in the communication scholarship, but there was no reflection here on the multiple problems with this approach."

Thank you for your suggestions concerning Section 6:

- As regards the need to strengthen the link with Section 3, we believe that the list of guiding criteria that will be included in Section 4 will also strengthen Section 6;

- As regards Figure 3, we will review the legend so that the comprehension is more immediate;

- As regards the survey, we will develop the reflection on the constraints related to this method, especially in the operational context of a communication campaign.

8)

"For me, the most interesting approach was the interviews, which provides the space for target audiences to explain what they understood, how it could help them or not, and yes, any contributions to resilience as defined in resilience assessment frameworks. This seems, to me, to be the best method for linking communication assessment to resilience assessment."

As for the Conclusions, we believe that these different methods are complementary and an assessment aimed to understand the impact of communication on resilience cannot rely on only one technique. We are also convinced that implementing advanced text mining and network representation of web communication contents in this field will innovate communication quality assessment. These techniques will be the object of our future research.

---

## Author Response (AR1)

**Answer to the Editor**

1)

*Editor: The authors have shown they will consider each of the reviewers points in the revision and in particular the flow, paragraph divisions, and sentences. I am therefore happy to give the opportunity to revise.*

Authors: We thank the Editor for this opportunity, for the time devoted to guide the review and for his helpful recommendations.
Besides considering each of the Referees' points, we have followed the Editor's advice to considerably improve the flow of the text, the paragraph divisions and the clarity of the sentences.

2)

*E: On numerous occasions, the authors say "additions" in their responses to the reviewers. I am slightly worried that the paper will become more confusing if only additions are made. As Reviewer 1 clearly stated, the paper needs considerable editing since it is confusing to read at present.*

A: We definitely agree with the Editor that too many additions to the manuscript would worsen the readability of the text. Hence we haven't added a 'Methodology' section and a 'Results and Discussions' section, as it was announced in our answer to Reviewer 1. Nevertheless, we have followed her/his recommendations to include further details on the methodology (in Sect. 5, named Sect. 6 in the first version of the manuscript) and to remove bibliographic references and references to the figures from the "Conclusions and perspectives".

**Answer to Reviewer 1**

A: We would like to thank Reviewer 1 for her/his careful review of our manuscript and her/his valuable comments and suggestions. We have addressed all of them and we believe that our manuscript is now much stronger. Here are our detailed responses to the issues raised by the Reviewer:

1)

*Reviewer 1: The paper asks some interesting questions, but fails to deliver. The writing is confusing and the arguments don't flow very well. Good framing of resilience and a fair amount of evidence from the literature.*

A: We appreciate that the Reviewer expressed her/his interest in the research

questions addressed in this paper.
We clarified what we want to deliver with respect to the objective of our research.
As it is stated in the "Introduction" (pp.1-2):

> "We propose to explore how urban resilience assessments can better take into account the interactions between science outreach and other resilience drivers. With this general scope, we examine:
> (i) The variables that are available in the context of a flood resilience project and that can be adopted as 'RCI' (Resilience Communication Indicators);
> (ii) The strengths and weaknesses of different methods that can be employed to monitor these indicators."

We have revised the "Conclusions and perspectives" (Sect. 6, p.16-17) so that the results delivered by this study are clearly expressed as well as its limits:

> "This study highlights that quantitative metrics are a promising tool for communication assessment in the framework of resilience strategies. The experiments carried out during the RainGain project have brought out valuable RCI.
> A preliminary study of Paris flood resilience strategies and the related communication plans has allowed us to identify five recurring categories of communication variables. Each category constitutes a helpful guidance to define RCI. At this stage of the research, we are cautious in generalising the validity of the RCI guiding criteria because they refer to the resilience communication strategies adopted in a unique region to cope with a specific climate risk. Nevertheless, this work paves the way to future developments. The same applies to the following conclusions that are the result of a limited number of small-scale experiments." [...]

> "The methods tested through these three experiments appear to be complementary and endorse the following conclusions: assessment aimed to investigate the impact of communication on resilience cannot rely on a unique technique and quantitative analysis is paramount in this context. Indeed, data in numerical form facilitate the study of interactions between the communication processes and other resilience drivers, such as meteorological events. Investigating these interactions is a necessary basis to integrate communication indicators in a wider urban resilience assessment."

Besides bringing to the fore, in the "Introduction" and in the "Conclusions and perspectives", what is the objective of the study and what is delivered, we have strengthened the link between the theoretical and the empirical parts, especially in the following sections:

I. In Sect. 3 we highlight that communication impact assessment is barely considered in the literature on urban resilience metrics and this requires further research on valuable communication indicators.

II. In Sect.4 we outline five guiding criteria to define relevant 'RCI' (Resilience Communication Indicators).

III. Section 5 specifies which RCI have been tested in the first two experiments (the "media monitoring" in Sect. 5.1 and the "survey" in Sect. 5.2) and why these indicators are useful to assess the impact of communication on urban resilience.

IV. Section 5 also clarifies that the "Interviews" experiment is not aimed at testing RCI but it explores a method that can be used for preliminary research or result validation.

2)

*R1: The methodology used is not clear at all, it's very vague. For example, there is no information on the sample used. For example, p13: "For this first evaluation a small sample was selected, since the main aim was to test a methodology that can be applied to different case studies. In order to perform a comparative experiment, a control group of respondents, who didn't attend the exhibition, was selected. " What is the sample size? How were the respondents identified and recruited? How many in the control group?*

A: As highlighted by the Reviewer, some details on the methodology were not specified or were only indicated in the figures and not in the main text. In the new version of the manuscript, Section 5 includes the following information on the methodology.

I. In Sect. 5.1 (pp.8-9) we describe how the data on news frequency and audience size have been collected for the "media monitoring experiment". We also specify the total number of news and the number of counted news per each type of media (TV, radio, printed press, digital press):

> " Among the communication values that have been monitored during the RainGain project, the media coverage has reached remarkable results that have far surpassed the target values. The data presented in this section have been collected from different sources:
> - Feedbacks from the Communication Department of École des Ponts that constantly monitors, through Europresse (europresse.com), if the media mention "École des Ponts".
> - Search on Google News of press news that include the key–word "RainGain".
> - Feedbacks from the researchers that were interviewed by the press on the RainGain project.
> - Data on the audience size of printed press have been collected on each newspaper website.
> From July 2011 to December 2015, we have counted a total number of 65 news on the RainGain project, published by the French, Dutch and Belgian press. These news include 29 articles on printed press, six TV reports, five radio reports, 25 Web news and Web Tv reports."

II. In Sect. 5.2 (p.12) we specify the size of the sample and of the control group in

the "survey experiment", we also describe the identification and recruitment processes:

> "37 respondents have been recruited on a voluntary basis among the 513 workers and 827 students of École des Ponts. They have been invited through internal mailing to complete an online survey form. [...]
> The survey included questions on the professional background of the respondents. These questions allowed to exclude six experts from the sample, in order to obtain a relative homogeneity in terms of background knowledge. As a result, the final sample consisted in 31 respondents. Other questions were aimed at identifying through which source of information the respondents learnt about the project. On the basis of these questions the sample has been divided in four subsets: 1) 13 visitors to the exhibition; 2) five visitors who also read the brochure distributed at the exhibition; 3) six respondents who received only informal information (from word of mouth); 4) 12 participants who never heard about the project. In order to perform a comparative experiment, the first subset has been considered as the experimental group with 13 respondents, while the third and fourth subsets have been considered as the control group with 18 respondents."

III. Section 5.3, at pp. 16-17, includes information on the number of respondents for the "Interviews experiment", and on the identification and recruitment processes:

> "While the video has been broadcast on Youtube and on the school screens, four interviews have been held. The questions aimed to appraise what kind of information the audience expected and how they interpreted the video contents. The respondents have been selected from the list of students invited to the conference and they have answered to the examiner on a voluntary basis. [...]
> A similar assessment, based on three open–ended questions (Table 3), has been undertaken in November 2015 to evaluate the impact of a workshop on RainGain (held during the Provin Climate Forum). The respondents were all the participants of the workshop: 20 pupils, aged eight years, who had been invited by the forum organisers. We chose the snapshot interviews (Fogg Rogers et al., 2015) as an investigation method because it is an alternative technique that is appropriate for a young audience and the context of a forum."

3)

*R1: Title - it would be good to clarify what exactly are the authors assessing. Is it the impact of outreach strategies? Something else?*

A: We understand the Reviewer's concern with the title. We have replaced it with "Assessing the impact of outreach strategies in cities coping with climate risks".

4)

*R1: The authors need to tidy-up the introduction a bit and make better use of paragraphs, for example. The text doesn't flow as well as it could if good punctuation, paragraphs, etc. was used.*

A: We also understand her/his concern about the text of the "Introduction" and we have revised the punctuation, paragraph divisions and those sentences that were unclear.

5)

*R1: Figure 1: this needs to be re-done. It's very hard to understand who much each bar accounts for as there's no scale. Also, you can't read the Y in full, for example "number of academic articles published based on" - based on what??*

A: We agree with the Reviewer that Figure 1 needed to be improved. We have replaced it with the following chart that includes the amounts concerning 2010-2012 (that were missing in the previous version of the manuscript), as well as complete and readable vertical axis titles.

[Figure]

**Figure 1: Monitoring of the frequency of RainGain communication activities in 2013. During the execution of the RainGain communication plan, the target values (to be attained by the end of the project) were periodically compared with the attained values.**

6)

*R1: Overall, all figures are confusing and difficult to interpret. The quality of the figures is not appropriate for publishing, they are more report-style.*

A: In order to improve the readability of the other figures, we have added some data labels, we have improved the axis titles, the legends and the captions.

7)

*R1: References are included in the conclusions section, as well as refs to figures - this is not appropriate for an academic journal.*

A: We thank the Reviewer for recalling us that references shouldn't be included in the conclusions. We have removed any bibliographic reference or references to the figures from the "Conclusions and perspectives".

8)

*Technical corrections: Overall the paper would benefit from being reviewed by a native English speaker*

We agree with the Reviewer concerning the need to benefit from a copy-editing service, which is a Copernicus standard service, since the authors aren't English native speakers.

**Answer to Reviewer 2**

A: The authors would like to thank the Referee for reviewing the manuscript and for providing the authors with her/his constructive remarks and recommendations.
We have considered all issues raised in her/his critique and we believe that we have improved our manuscript after addressing these valuable comments. Here is a list of our responses to her/his comments and we hope that these changes will satisfy her/his requirements:

1)

*Reviewer 2: This manuscript set out to assess how science communication can contribute to urban resilience, using empirical studies to demonstrate how*

*communication can increase scientific impact on resilience according to one or more resilience assessment frameworks. This is promising goal, and a potentially important contribution to the scholarship; enriching these frameworks with a more carefully consideration for communication. But, excepting a short discussion in the conclusion, unfortunately the manuscript fails to properly establish this link. It looses its way, its significance, and the reader. In my opinion this paper could be publishable subject to major revisions around its original purpose.*

A: We appreciate the Reviewer's acknowledgement for the "promising goal" of our research. We agree that the link between resilient metrics and communication impact needed to be made more explicit. We have therefore strengthened this connection in all the Sections of the new version of the manuscript:

I. In the "Introduction", at p. 2, we state that:

> " We consider that [communication] impact is not sufficiently explored in the literature on urban resilience indicators, despite the growing importance of science outreach in urban resilience projects and strategies."

II. In Section 2 we observe that the theoretical frame of social-ecological resilience is particularly suited to appraising communication impact on urban resilience. We also discuss that turning social-ecological resilience theory into practice requires advances in public outreach and citizens' engagement. We go into detail on this point (with extracts from the manuscript) at pp. 10-11 of this document (answer 4 to Reviewer 2).

III. In Sect. 3 we argue, with references to the literature, that communication impact indicators are barely considered in the current resilience assessment research and practices:

> "According to Charrière et al. (2017), impact assessment of risk communication campaigns isn't a widespread practice yet. This trend can be also observed in the literature on resilience indicators. This section presents three resilience assessment frameworks that consider the impact of communication processes, a feature that is not so common among the available indicators for cities coping with weather extremes (for a review, see Vicari et al., 2015). [...]
>
> Among the nine assessment frameworks reviewed by Vicari et al. (2015), only the three methods presented in this section refer to communication effects. Furthermore, the RA, DROP-BRIC and HFA methods offer a range of communication indicators that aren't enough sophisticated to evaluate science outreach activities, especially in terms of quality."

IV. In Sect. 4 we outline five categories of communication variables. These categories can serve as guiding criteria to define relevant 'RCI' (Resilience Communication Indicators). We then specify which guiding criteria have been

used to identify the communication variables that are the object of the experiments presented in Sect. 5.

> "A hypothesis of relevant communication variables has been outlined by Vicari et al. (2016). In this former study, quantitative variables are selected on the basis of the communication objectives, target audiences and communication actions of 13 flood resilience strategies, implemented in Paris from 2003 to 2017. These variables are conceived as tools that can be adopted by the decision makers to evaluate if the communication goals have been achieved. Hence, these indicators are tailored to each resilience strategy and context specificity and they rely on the available communication data that can be collected for evaluation. Nevertheless, these variables can be grouped into five recurring categories that are listed below. The following categories can serve as guiding criteria to include relevant RCI in a wider urban resilience assessment (such as those presented in Sect. 3). [...]
>
> The next section presents three different kinds of experiments that have been carried out in the framework of the RainGain project. Each experiment takes into account some of the five guiding criteria listed above. More specifically, the first experiment "Media coverage monitoring" explores the intensity of communication (criterion i), it compares different time periods (criterion iv) and highlights the correlations between communication and another resilience driver, i.e. a meteorological event (criterion v). The second experiment "Survey administered to the visitors of an exhibition" explores the quality of communication (criterion ii) and compares different sub-groups of audiences (criterion iv). The third experiment "Interviews" concerns the quality of communication (criterion ii)."

V. In Section 5 we specify which RCI have been tested in the first two experiments (the "media monitoring" in Sect. 5.1 and the "survey" in Sect. 5.2) and why these indicators are useful to assess the impact of communication on urban resilience. We also clarify that the "Interviews" experiment is not aimed at testing RCI but it explores a method that can be used for preliminary research or result validation. For further details on this point, see answer 9 to Reviewer 2 at pp.13-14 of this document.

VI. In the "Conclusions and perspectives" we recall that there is a need for valuable communication metrics to implement urban resilience. We also stress out that, despite the limits of small-scale experiments, these have allowed to test relevant RCI:

> "The increasing awareness of the role that citizens can play as active actors of urban resilience make essential the development of relevant communication indicators. This study highlights that quantitative metrics are a promising tool for communication assessment in the framework of resilience strategies. The experiments carried out during the RainGain project have brought out valuable RCI. [...]
>
> The methods tested through these three experiments appear to be complementary and endorse the following conclusions: assessment aimed to investigate the impact of communication on resilience cannot rely on a unique technique and quantitative analysis is paramount in this context. Indeed, data in numerical form facilitate the

study of interactions between the communication processes and other resilience drivers, such as meteorological events. Investigating these interactions is a necessary basis to integrate communication indicators in a wider urban resilience assessment."

2)

*R2: There are two words here that jar with science communication scholarship. 'Progress' to resilience is sticky: when resilience is a moving target, progress is too. 'Objectively' is similarly sticky. Given communication is often a two-way communication, the communicator is actively embedded in the communication act, and it is then difficult for them to objectively 'step outside' of this communication. Suffice to assess, without emphasising the objectivity.*

A: As recommended by the Reviewer, the word "objectively" and "progress" have been removed from the Abstract as they can be seen as controversial. However, we would like to recall our intent to move from a theoretical to a practical approach to social-ecological resilience. As it is mentioned in the "Introduction", Holling, Folke and others emphasise that renewal, development and adaptation are essential characteristics of a resilient system. But this does not prevent to assess resilience. Indeed, as specified in Sect. 2, once it is clear what is the system and what is the disturbance we are interested in, it is possible to establish resilience metrics and compare the corresponding variables in different systems (e.g. different cities) or within the same system at two different moments.

> Introduction, p.1: "The 'social–ecological resilience' approach outlined by Holling (1973) departed from the mainstream interpretation of resilience by pointing at renewal, re–organisation, innovation, development and adaptation as important capacities of a resilient system (Gunderson and Holling, 2002; Berkes et al., 2003; Adger, 2006, Folke et al., 2010). This approach presupposes the use of resilience indicators as an empirical basis to translate the concept of social-ecological resilience into practice."

> Sect.2, p.3: " A first necessary step to design resilience metrics is to identify the disturbance and the system we are interested in. Even though the interplay with other scales and other shocks or stresses shouldn't be ignored, answering the question "resilience of what, to what?" (Carpenter et al., 2001) is an essential basis to establish resilience indicators. The same relevant variables can be then compared in different systems (e.g. different cities) or in the same system at different moments."

3)

*R2: The introduction...is not an introduction. It does not introduce the work that will be done in the paper, including a clear statement of the research question or aim. This material can be moved to the background Section 2.*

A: As it was recommended by the Reviewer, we have specified in the

"Introduction" (p.2) what are the rationale, the research questions, and the purpose of this research:

> "We consider that [communication] impact is not sufficiently explored in the literature on urban resilience indicators, despite the growing importance of science outreach in urban resilience projects and strategies."

> "We propose to explore how urban resilience assessments can better take into account the interactions between science outreach and other resilience drivers. With this general scope, we examine:
> (i) the variables that are available in the context of a flood resilience project and that can be adopted as 'RCI' (Resilience Communication Indicators);
> (ii) the strengths and weaknesses of different methods that can be employed to monitor these indicators."

As suggested by the Reviewer, most of the information included in the first version of the "Introduction" has been condensed and moved to Sect. 2, at p. 2:

> "According to the social–ecological resilience perspective, 'resilience' can be defined as the "the capacity of a system to absorb disturbance and reorganize while undergoing change so as to still retain essentially the same function, structure, identity, and feedbacks" (Walker et al., 2004). 'Transformability' and 'adaptability' (Folke et al., 2010) are considered as essential characteristics of a resilient system. This approach puts the accent on uncertainty, non–linear dynamics, interplay between gradual change and rapid change (Walker and Meyers, 2004). The trajectory followed by a system after a perturbation can't be described as reaching 'stable states' or 'equilibriums', but rather with the concepts of 'regimes' or 'attractors' (Carpenter, 2003).
> These dynamics involve interactions across different time and space scales (Schertzer and Lovejoy, 2004; Tchiguirinskaia et al., 2014)"

4)

*R2: Section 2: interesting, but there is no mention of science communication. The focus is on resilience assessment metrics alone.*

A: We have followed the recommendation of the Reviewer to improve Sect. 2 by making more explicit the link with science communication. We state that the theoretical framework of social-ecological resilience allows investigating the interplay between communication processes and other social-ecological or physical environmental factors in the city. We also explain that one of the concrete consequences of resilience implementation is that citizens' perceptions and public outreach campaigns have considerably gained in importance.

> "[The social-ecological resilience approach offers a] multi-dimensional perspective [that] is particularly suited to studying the complexity of urban systems and the influence of communication processes on resilience. Cities have multiple components and functions, including the communication factors that can be defined as part of the social dimension and that obviously have interdependencies

with the economic, physical and environmental dimensions. [...]

Putting the concept of social–ecological resilience into practice involves relevant changes in policy and decision–making. Indeed, the social–ecological resilience approach emphasizes the need to apply the principle of subsidiarity, i.e. to decentralise risk management, to encourage citizen participation and share responsibilities with them (Tanguy, 2015). In Sect. 4, we discuss some cases of recent resilience strategies implemented in the Paris region that entail public engagement activities. These examples illustrate how public outreach and citizens' perceptions are gaining importance, as a consequence of the implementation of the subsidiarity principle."

5)

*R2: I missed some up-front reflections on all of the unquantifiable elements of resilience. This kind of quantitative work seems, to me, to draw a very narrow concept of resilience.*

A: We understand the Referee's concern about the unquantifiable aspects of resilience. Though, we have kept the focus on quantitative assessment, since it facilitates an analysis of the interactions between the socio-economic factors, such as communication, and the physical environmental factors. Qualitative assessments give valuable insights on resilience, but for our research purposes interviews or focus groups are more suited to preliminary studies or result validation.
To ensure that this premise of our research is clearly stated we made the following additions:

Introduction, p. 2: "These indicators are based on quantitative variables since numerical data allow exploring the correlations between communication and other resilience drivers."

Sect. 3, p. 5: " We should finally note that the RA approach employs quantitative variables to study the correlations between social factors and ecological factors. Moreover, the DROP-BRIC method shows that quantitative indicators facilitate the comparison of different spatial scales. These observations are in line with the conclusions of the review by Vicari et al. (2015) that have led us to focus on quantitative communication variables, as those presented in the next section."

Sect. 5.2, p. 12: "However, rather than to obtaining results that could be generalised to a wider population, our main objective was to test if quantitative research can be employed to evaluate the quality of communication. Indeed, this method and the research technique presented in the previous experiment have a common characteristic: they provide numerical data that are adequate to integrate communication assessment in a wider urban resilience assessment."

6)

*R2: Section 3 is unnecessarily long and doesn't contribute much to this papers discussion. It is enough to note that there are many different frameworks available, and which frameworks this manuscript focused to evaluate the impact of science communication. It could be shortened and combined with Section 2.*

A: We have followed the advice of the Reviewer to shorten Sect. 3 by half, by making more explicit the link with science communication, and by strengthening the connection with the empirical part. Indeed, instead of mentioning 9 resilience assessment frameworks (as in the first version of the manuscript) we describe only three of them that refer to communication processes. In our view, these resilience metrics don't accurately investigate communication processes. Hence in Sect. 4 we outline five guidelines to innovate these metrics.

7)

*R2: Section 4 is interesting for this paper; to me it is the core. It should provide more detail on the mechanisms between communication and resilience, to help us better understand the assessment in terms of influence, rather than as bald measurements that reveal no clear causal link between the communication and resilience. I expected to see how communication assessment linked to the metrics of one or more resilience assessment frameworks, but it was missing.*

A: We understand the Reviewer concern with Section 4. In the new version of Sect. 4, we stress out that communication is a key driver of resilience, and we support this assertion with some examples of resilience strategies from the Paris region. We also establish a link between communication assessment and the resilience assessment methods presented in Sect. 3. Indeed, Sect. 4 outlines five guiding criteria to define valuable RCI that can be included in wider resilience assessments (such as those presented in Sect.3), and thus innovate them.

8)

*R2: Some of the reflections from the conclusion could be moved up here.*

A: We thank the Reviewer for this suggestion. We have removed the list of characteristics of relevant communication indicators from the "Conclusions" (p. 19 of the first manuscript version). Indeed, this topic is thoroughly developed in the new version of Sect. 4 that lists guiding criteria to define RCI.
We have also moved the references to "Iterative Risk Mangement" (Keating et al., 2014), the "Urban Weather Sensing Lab in Amsterdam" (Koole et al., 2015) from the "Conclusions" to Sect.4. As it is stated under the third criteria "Participatory communication" at p. 6, this kind of experience constitute a

potential source of data on participatory communication:

> "Data on public engagement can be easily collected in the case of projects involving the use of social media (Grandi and Neri, 2014; Topping and Illingworth, 2016) or mobile app for 'citizen science' (Keating et al., 2014; Koole et al., 2015)."

8)

*R2: Section 5 does not add much. It should be linked to the manuscripts core work, or removed.*

A: We agree with the Reviewer concerning Section 5. This has been removed since it's not relevant to answer the research questions.

9)

*R2: Section 6 fails to make any link to resilience assessment metrics. It reads independent to the rest of the paper, as an account of three different communication assessment approaches for a resilience project. I could see how this communication assessment contributed to resilience as framed in a resilience assessment framework.*

A: Thank you for your suggestions concerning Section 6 (named Sect. 5 in the new version of the manuscript). As regards the need to strengthen the link with the resilience assessments presented in Sect. 3, we believe that the list of guiding criteria that have been included in Section 4 has also strengthened Sect. 5. Section 4 specifies that each experiment is based on one or two guiding criteria (see answer 1/IV to Reviewer 2 at pp. 8 for the corresponding extract). Section 5 indicates which communication variables are tested in the first two experiments and why these are useful to evaluate the impact of communication on urban resilience. It also specifies that the third experiment is not aimed at implementing RCI but it tests a method that is valuable for preliminary research and result validation.

> Sect. 5.1, p.10: "The frequency of press news and the audience size are two RCI that allow identifying the population that has received a specific message. This is a necessary step to evaluate the communication effects on citizens' perceptions and urban community resilience. The RCI employed in this experiment also allow to observe how the resonance of a message evolves over time (Fig. 2 and 3) and to identify possible correlations with other resilience drivers (e.g. a meteorological event, as it shown in Fig. 1)."

> Sect. 5.2, p.10: "The experiment presented in this section illustrates how RCI based on a survey can capture the quality of communication and if the audience has understood and accepted a message. Indeed, even if a communication activity reaches a wide public, the communication impact on urban resilience will vary according to the way the message is perceived. Survey questions, such as those

presented in this experiment, provide variables (e.g. frequency of correct questions, frequency of high risk perception) that can be used as RCI to assess the respondent comprehension and perception."

Sect. 5.3, pp. 14-15: "These research techniques don't provide quantitative variables that used as RCI. Nevertheless, this is a helpful evaluation method to be adopted for exploratory studies or to validate the results of a survey."

10)

*R2: To the three methods, the quantitative work I understood, apart from Figure 3. Of course its limited in what it shows, but the authors admit that.*

A: We thank the Reviewer for highlighting that Figure 3 is not easy to understand. We have replaced it with the following chart where we have added titles to the axis and we have improved the caption so that the comprehension is more immediate:

[Figure]

**Figure 3: The ratio between the number of articles and the audience size of printed press. The differences between the two curves are due to the fact that different newspapers have different impacts in term of audience size, hence the impact of a news is variable according to the newspaper that publishes it.**

11)

*R2: The qualitative assessment is more problematic, as presented now. These kinds of 'science comprehension tests' have been around a long time in the communication scholarship, but there was no reflection here on the multiple problems with this approach.*

A: As regards the survey, we have added a reference to budgetary and time constraints related to this method. Furthermore, we state that surveys provide only aggregated analysis and cannot give details on the respondent individual perspective or on the cognitive processes that shape his perceptions. Nevertheless, this method appears to be valuable in the context of this research because it provides numerical data that facilitate the comparison between communication processes and other resilience drivers:

> Sect. 5.2, p. 12: "The sample has been expected to be small, since no monetary incentive was provided for survey participation and there was no examiner who could individually reach each potential respondent to solicit his answers. We were also aware that surveys give limited insights on the cognitive processes that shape individual and social perceptions. However, rather than to obtaining results that can be generalised to a wider population, our main objective was to test if quantitative research can be employed to evaluate the quality of communication. Indeed, this method and the research technique presented in the previous experiment have a common characteristic: they provide numerical data that are adequate to integrate communication assessment in a wider urban resilience assessment."

12)

*For me, the most interesting approach was the interviews, which provides the space for target audiences to explain what they understood, how it could help them or not, and yes, any contributions to resilience as defined in resilience assessment frameworks. This seems, to me, to be the best method for linking communication assessment to resilience assessment.*

As it is stated at p. 18 in the "Conclusions and perspectives" (see the following extract), this study shows that different research methods, presented in the empirical part of the paper, are complementary. An assessment aimed at appraising the impact of communication on resilience cannot rely on a unique technique. Furthermore, quantitative methods are more suited to establishing RCI, but qualitative research, such as interviews, can be employed for preliminary studies and result validation.

[revised manuscript text omitted]

rosa 28/8/18 00:19

rosa 29/8/18 11:21

rosa 29/8/18 11:21

rosa 31/8/18 19:11

rosa 3/9/18 01:06

rosa 30/8/18 16:21

rosa 30/8/18 16:25

rosa 28/8/18 17:15

rosa 17/7/18 11:43
Moved (insertion) [1]

rosa 3/9/18 01:11

rosa 27/8/18 00:15

rosa 3/9/18 01:06

rosa 3/9/18 01:11

rosa 3/9/18 01:12
Deleted: ByBecause of the complexity of systems, that continuously evolve and are characterized by multiple interactions across spatial and temporal scales, the trajectory followed by a system after a perturbation cannot be described with the help of the concept of stable states or equilibriums, but rather with the concepts of regimes or attractors (Carpenter, 2003).   … [21]

rosa 17/7/18 11:43
Moved up [1]: 2 Implementing social–ecological resilience .

rosa 1/9/18 15:39

rosa2 3/9/18 01:06

public engagement activities. These examples illustrate how public outreach and citizens' perceptions are gaining importance, as a consequence of the implementation of the subsidiarity principle.

Going beyond theory and implementing resilience require resilience metrics: relevant indexes allow decision makers to compare the costs of resilience enhancement actions with the economic, environmental, social, and sanitary costs of non‑action. Resilience metrics also help to set up clear objectives at the beginning of a project, to evaluate and improve management capacities, to increase transparency and stakeholders' involvement during and after a project. According to Carpenter et al. (2001), resilience metrics allow testing hypotheses on the dynamics of systems and enable cross‑system comparisons.

A first necessary step to design resilience metrics is to identify the disturbance and the system we are interested in. Even though the interplay with other scales and other shocks or stresses shouldn't be ignored, answering the question "resilience of what, to what?" (Carpenter et al., 2001) is an essential basis to establish resilience indicators. The same relevant variables can be then compared in different systems (e.g. different cities) or in the same system at different moments. In this paper, the focus is on cities facing climate risks: in the next sections we present examples of resilience assessment frameworks that are adequate to urban areas coping with extreme weather (Sect. 3); we then discuss the role of communication in flood resilience strategies implemented in the Paris region and we outline guidelines to define RCI for cities facing climate risks (Sect. 4); we finally compare different communication assessment techniques that have been tested in the framework of RainGain, a European research project on urban flood resilience (Sect. 5).

The resilience assessment approaches presented in the next section are quite heterogeneous in terms of the concept of resilience they refer to, the system and disturbances they consider, the selection of indicators and variables, the degree of on‑site implementation. However, none of these approaches sufficiently investigates the impact of communication processes and, more specifically, of science outreach.

**3 Communication indicators in the literature on resilience assessment techniques**

According to Charrière et al. (2017), impact assessment of risk communication campaigns isn't a widespread practice yet. This trend can be also observed in the literature on resilience indicators. This section presents three resilience assessment frameworks that consider the impact of communication processes, a feature that is not so common among the available indicators for cities coping with weather extremes (for a review, see Vicari et al., 2015).

**3.1 Resilience Alliance**

'Resilience Alliance' (RA) (Resilience Alliance, 2010) is an international, multidisciplinary research organisation that develops guidelines to assess resilience of social‑ecological systems and to implement sustainable development strategies. RA outlines an assessment framework that is consistent with the social‑ecological approach. According to this method, multiple spatial and temporal interacting scales most be considered. Furthermore, for each variable, a threshold

rosa 26/8/18 21:33
Deleted: Since 2014 in the Paris region, public Another relevant aspect specifically concerns urban resilience to climate change: the social–ecological resilience approach puts the focus on adaptation and mitigation to climate change, rather than on the need to reduce greenhouse gases emissions and urbanization authorities have started to embrace the principle of subsidiarity through the recent flood resilience strategies. In this context,…ow ci ... [22]

rosa 28/8/18 17:22

rosa 26/8/18 21:37

rosa 17/7/18 13:49

rosa2 3/9/18 01:06

rosa 17/7/18 17:14

rosa 26/8/18 19:29

rosa 17/7/18 13:53

rosa2 3/9/18 01:06

rosa 19/7/18 00:26

rosa 3/9/18 01:15

rosa 27/8/18 11:38
Moved (insertion) [6]

rosa 27/8/18 11:42
Deleted: The resilience indicators presented in this section are drawn from the report "Review of alternative approaches to assess resilience to extreme weather" (Vicari et al., 2015). Three assessment frameworks have been selected from the review since they refer to resilience drivers that are related to communication processes.

rosa 18/7/18 18:36
Moved (insertion) [2]

rosa 3/9/18 01:15

[revised manuscript text omitted]

**"If I work on the campus, exposure to the radar frequencies is:"**

| | Intense | Moderate | Very moderate | Non-existent |
|---|---|---|---|---|
| All respondents (31) | 10% | 23% | 42% | 25% |
| 13 visitors to the exhibition (experimental group) | 9% | 10% | 45% | 36% |
| 5 visitors to the exhibition who also read the brochure | | 20% | 40% | 40% |
| 18 respondents who didn't see the exhibition (control group) | 11% | 33% | 39% | 17% |
| 12 respondents who never heard about the project | 8% | 42% | 33% | 17% |
| 6 respondents who recieved only informal information (word of mouth) | 17% | 17% | 50% | 16% |

**Figure 5:** Answers to a survey question evaluating the risk perception of the visitors after the RainGain exhibition.

**5.3 Interviews**

While surveys with close-ended questions allow quantifying the results, interviews can reveal more insights on the reactions and reasoning of the respondents. These research techniques don't provide quantitative variables that can be used as RCI. Nevertheless, this is a helpful evaluation method to be adopted for exploratory studies or to validate the results of a survey.
* * *
[2] We have computed a 2x2 contingency table, for each survey question, with the frequencies of: a) the correct answers or wrong answers; b) the correct answers of the control group; c) the wrong answers of the experimental group; d) the wrong answers of the control group. We have then applied the Fisher's Exact test because in all the 2x2 contingency tables at least one value is N ≤ 5. The test uses the following formula where the 'a,' 'b,' 'c' and 'd' are the individual frequencies of the 2X2 contingency table, and 'N' is the total frequency:

$$p = ((a+b)!\,(c+d)!\,(a+c)!\,(b+d)!)\,/\,a!\,b!\,c!\,d!\,N!$$

rosa2 3/9/18 01:06
rosa 31/8/18 14:27
rosa2 3/9/18 01:06
rosa 31/8/18 14:28
rosa2 3/9/18 01:06
rosa 27/8/18 19:38
rosa 28/8/18 19:27
rosa 10/8/18 17:42
rosa2 3/9/18 01:06
rosa 16/8/18 01:06
rosa 15/8/18 23:47
rosa 3/9/18 01:06
Formatted [109]
rosa 2/9/18 19:33
rosa2 3/9/18 01:06
Formatted [110]
rosa 27/8/18 19:45
Formatted [111]
rosa2 3/9/18 01:06
Formatted [112]
rosa 27/8/18 19:45
Formatted [113]
rosa2 3/9/18 01:06
Formatted [114]
rosa 3/9/18 01:06
Formatted [115]
rosa2 3/9/18 01:06
Formatted [116]
rosa 3/9/18 01:06
Formatted [117]
rosa2 3/9/18 01:06
Formatted [118]
rosa 27/8/18 19:45
Formatted [119]
rosa2 3/9/18 01:06
Formatted [120]
rosa 27/8/18 19:45
Formatted [121]
rosa 27/8/18 19:45
Formatted [122]

[revised manuscript text omitted]

rosa2 3/9/18 01:06

rosa 1/9/18 19:32

rosa 10/8/18 19:16
**Formatted** ... [160]

rosa2 3/9/18 01:06

rosa 1/9/18 19:41

rosa 1/9/18 19:39

rosa 10/8/18 19:16
**Formatted** ... [163]

rosa2 3/9/18 01:06

rosa 10/8/18 19:16
**Formatted** ... [162]

rosa 10/8/18 19:16
**Formatted** ... [164]

rosa 1/9/18 19:43

rosa2 3/9/18 01:06

rosa 10/8/18 19:16
**Formatted** ... [165]

rosa 1/9/18 19:43

rosa 10/8/18 19:16
**Formatted** ... [167]

rosa 15/8/18 23:48

rosa2 3/9/18 01:06
**Formatted** ... [169]

rosa2 3/9/18 01:06
**Formatted** ... [170]

rosa2 3/9/18 01:06
**Formatted Table** ... [171]

rosa2 3/9/18 01:06
**Formatted** ... [172]

rosa2 3/9/18 01:06
**Formatted** ... [173]

rosa 2/9/18 19:48

rosa2 3/9/18 01:06

rosa 10/8/18 19:30
**Formatted** ... [174]

rosa2 3/9/18 01:06
**Formatted** ... [175]

rosa 10/8/18 19:32
... [176]

rosa 16/8/18 14:50

rosa 30/8/18 22:54
**Formatted** ... [177]

rosa 13/8/18 15:13
**Moved (insertion) [3]** ... [178]

rosa 28/8/18 00:02
... [179]

rosa 3/9/18 01:06
**Formatted** ... [180]

rosa 27/8/18 23:51
... [181]

rosa 13/8/18 16:35
**Moved (insertion) [4]** ... [182]

rosa 27/8/18 23:54
... [183]

[revised manuscript text omitted]

---

## Author Response (AR2)

First the authors would like to thank the Editor and the Reviewers for their time and comments that helped improve the manuscript. We have addressed all the issues raised by the Reviewers and the Editor. Hopefully the changes implemented will satisfy their requirements!

ANSWER TO THE EDITOR

*Editor: This reviewer (#3) has provided a thorough list of elements that they believe will improve your paper. Reviewer #2 is happy with the changes you made in the last round, but requests a thorough check of language before final submission.*
*I would like to invite you to consider Reviewer #3's suggestions and to ensure that the english is professionally checked, at the very least by a native english speaker or expert.*

Authors: We thank the Editor for guiding the review. We agree that Reviewer 3 provided relevant suggestions. We have addressed all of them, as it is detailed in the answers to Reviewer 3, and we have improved the paper accordingly.  Furthermore, the manuscript has been reviewed by an English expert.

ANSWER TO REVIEWER 2

*Reviewer 2 (Report #1): I congratulate the authors for having transformed the manuscript into a robust and interesting contribution to the climate resilience literature. They have addressed all of my concerns; strengthening the argument, and giving the paper a coherence and completeness that is satisfying to read. They responded well to my concerns about quantitative assessment, outlining the need for a common currency when interfacing with other quantitative resilience assessment frameworks. I am left with just one minor technical concern about the English grammar in some places, which distracts from what is otherwise an interesting paper. The manuscript would benefit from one quick final review of the English prior to publication.*

Authors: We would like to thank the Reviewer for his feedback and we are happy to see that he appreciated the improvements that we have made in the second version of the paper and that we satisfied his requests. We agree with him that there is a need for a common currency to develop an interface with other quantitative resilience indicators. Following his comment about the English grammar, we have asked to an English expert to review the paper.

ANSWERS TO REVIEWER 3

*1)*

*Reviewer 3: This paper aims to appraise how public outreach campaigns influence urban resilience in a specific case study around flooding in Paris, France. Whilst this*

*idea is commendable and metrics to help evaluate the impact of public outreach can influence urban resilience are a good idea, this paper struggles to connect the work done with the broader aims proposed in the introduction. As such I think this paper should be resubmitted after major revisions.*

Authors: We are pleased that Reviewer 3 finds interesting our research aim and we thank him for giving us relevant suggestions. We believe that, thanks to his comments and the modifications (that are detailed below), the link between the research goal and the presented work has been strengthened.

2)

*R3: Many of the issues of this paper I feel are down to problems of structure - that with a more logical structure and careful selection of data this paper could make its case for using qualitative data as a metric for assessing resilience changes more effectively. Having said that I think there is also a bigger issue that would need to be addressed in a point raised by one of the previous reviewers, on the usefulness of using only quantitative data in a study like this - that the limitations of such approaches need to be seriously addressed as they are central to any discussions of resilience in urban landscapes.*

A: We understand the concerns of the Reviewer about the following issues:
a) The logical structure: we agree that the paper would benefit from a clear description of the logic that we have followed to select relevant metrics.

b) The selection of data: we are convinced that all the experiments provide valuable data, even though we recognise some limits that are discussed in the paper (see Answer 3). We agree that a supplement with the complete answers by the participants and demographic information are useful additions to the paper (see Answer 9).

c) Qualitative and quantitative data: we believe that both kinds of data have their advantages and limits. As it is highlighted in the new version of the manuscript, we acknowledge the value of qualitative research and the limits of quantitative data. Nevertheless, we use quantitative variables as RCI, since quantitative data facilitate the identification of space-time variability and of correlations between communication processes and other physical environmental processes. These correlations allow to integrate communication assessment in a wider resilience assessment. We also prefer to maintain this point of view because it is in line with the observation by Reviewer 2 (in Report #1) about 'a need for a common currency when interfacing with other quantitative resilience assessment frameworks'.

In order to put emphasis on point a) and c), we have improved the text with several additions and modifications in Sect. 3.4 (p. 5 l. 15-27 and p. 6 l. 3-24), Sect. 4 (p. 7 l. 10-21), the Conclusions (p. 18 l. 6-8 and l. 18-21; p. 19 l. 1-3).

3)

*R3: Finally I think the data presented in section 5 (5.1, 5.2 and 5.3) was used as the basis for several large leaps of reasoning, and I would question if the data collection methods presented were actually measuring what the researchers described?*

A: We understand the Reviewer's concern about Sect. 5. We hope that that the modifications, detailed in the next answers, will bring forward the legitimacy of our observations. Nevertheless, we recognise that the conclusions based on these small-scale experiments would need further validation with larger samples, as we state in the Conclusions (p. 17 l. 27 - p. 18 l. 4). Furthermore, we have made explicit through modifications to Sect. 5.2 (in the captions of the Fig. 3-4 and in the main text at p.14-15) that the percentages, describing the results of the questionnaire, refer to the group of respondents and not to a larger population.

4)

*R3: This was most striking to me in sections 5.2 where there were several variables not identified (such as how long had it been since the exhibition, did the participants engage with any other sources of information in addition to the exhibition, did any of the participants have experience of the topic other than professional? Also several factors were not explained in anywhere enough detail (respondents had heard about the project, but from where?).*

A: Following the Reviewer's comment, we have included the requested information in Section 5.2 (p. 12 l. 6-10) and in Footnote 3 (p.12).

5)

*R3: In section 5.3 a lot of the problems seemed to be a result of the design of the survey and as mentioned by reviewer two previously, a reflection on the vast body of work on surveys to establish science literacy and comprehension in the context of your own work here would be very useful.*

A: We agree with the Reviewer that the paper would benefit of references to the scientific literature on questionnaire and interview techniques, and in particular as tools to investigate science outreach and risk communication. We have added in Sect. 3.4 (p. 5 l. 28 - p. 6 l. 20) these references that have allowed us to better frame the experiments.

6)

*R3: Additionally I think it is dangerous to assume that just because a method provides numerical metric that it is a valuable way to measure the outcome of public outreach.*

A: As it is highlighted by this comment of the Reviewer, it might be unclear that we don't consider numerical metrics as the most valuable in general. On the contrary, we consider that qualitative and quantitative research techniques are complementary. Nevertheless, in the context of resilience assessment, we prefer to base RCI on quantitative variables to facilitate an integrated analysis of communication processes and physical environmental processes and an investigation of space-time variability. With this perspective in mind, we consider qualitative assessment as extremely valuable for exploratory studies. We have put more emphasis on these key points of our research in Sect 3.4 (p. 5 l. 15-20; p. 6 l. 17-20) and in the Conclusions (p.19 l. 1-5).

7)

*R3: One of the key structural elements I think needs to be addressed would be a clear explanation of the RCI, they appear to be introduced early, before the body of literature, but it is unclear if these measure are the result of a previous study or a novel concept by the authors, ...*

A: Following the comment of the Reviewer, in the third version of the paper, we have stressed out that RCI is a novel concept introduced by a former study carried out by the authors: Introduction (p. 2 l. 9-11), Sect. 4 (p.7 l. 10-22).

8)

*R3: ...followed by more depth of data presented in the results sections, perhaps in tabular form? If not in the paper itself then basic demographic data for the participants should be available in a supplement to validate the quality of the data you collected.*

A: We thank the Reviewer for these helpful suggestions. We have added Supplement 1 with the complete list of questions and answers rates to the questionnaire and demographic data on the respondents, and Supplement 2 the complete answers to the snapshot interviews. Unfortunately we don't have the authorisation from the respondents to publish their uncut answers to the interview on the video. We have also included demographic data on the participants to the interviews in Sect. 5.3 (p. 16 l. 17-19 and p.17 l. 19-20).

9)

*R3: Additionally I would question if the case study provided in this paper actually does what the title suggests and assess the impact of the public outreach campaigns in this case in Paris. There is no data shown to measure a change in any resilience metrics and no clear measures of 'success' were presented. In fact this paper seems to be more*

*of an assessment of outreach strategies used in a situation where resilience was an issue, than an assessment of how those strategies impacted resilience of participants. Perhaps a slight reframing of the paper may help clarify this point, or if the authors have additional supporting data that placed the results of the communications study in the context of other local resilience measures, that would be great.*

A: We understand the doubts raised by the Reviewer concerning the link between resilience assessment and communication assessment. In the new version of the manuscript we have stressed out the following points:
1) The aim of the paper is to focus on communication impacts in the framework of resilience assessment. A general resilience assessment of an urban area wouldn't have left enough space to discuss, in detail, communication impacts.
2) For this reason, we first assume that communication enhances resilience if it contributes to achieving the goals of pre-defined resilience strategies (e.g. the strategies reviewed by Vicari et al. (2016) or the RainGain project); we then detail in Sect. 5.1 and 5.2 how press relations and the exhibition have contributed to achieving the project goals, i.e. to enhance local resilience to flood.
3) We have defined which variables allow monitoring communication impact on resilience through a former review of 12 resilience strategies implemented in Paris, their communication goals, activities, target audiences.

We have put emphasis on these three points with additions and modifications in Sect. 3.4 (p. 5 l. 23-27; p. 6 l. 3-12), Sect. 4 (p. 7 l. 10-21), Sect. 5.1 (p.11 l. 11-13), Sect. 5.2 (p.15 l. 4-12).

*R3: Additional points:*

*On reflection of the previous reviewers comments:*

10)

*R3: I agree with reviewer one, this paper needs to be reviewed by a native English speaker before publication.*

A: We have followed the recommendation of the Reviewers, the new version of the manuscript has been reviewed by an English expert.

11)

*R3: Review one, point 3 - I would like to see more evidence of this - what are the impact of the outreach strategies?*

We agree with the Reviewer that this point should be brought forward. To reply to this comment we refer to Answer 9. In the specific case of the experiments, we have

observed the communication impacts described in the extracts from Sect. 5.1 and 5.2.

12)
*R3: Reviewer two, point 2 - I think the authors could provide more context here - I think there is a very valuable point made by reviewer two on the place of the researcher in the data and I would like to see this addressed.*

A: Following the request of the Reviewer, we have included new details on the context of the evaluation process in Sect. 5 (p. 8 l. 26-33).

13)

*R3: Reviewer two, point 7 - I agree with this point, but in your answer you state that you draw a line between communications and resilience assessment, but I would ask is that then followed up by your data?*

A: We understand the Reviewer's concern about how our data show the impact of communication on resilience. We believe that the modifications described in Answers 9 and 11 will satisfy his request for clarification. Furthermore, we should recall that four of the five RCI defined in Sect. 4 are then tested through the experiments presented in Sect. 5. We have stressed out this point with additions and modifications to Sect. 4 (p.8 l. 7-13), Sect. 5.1 (p.11 l. 7-14), Sect. 5.2 (p.11 l. 16 - p.12 l. 3; p.15 l. 4-12).

14)

*R3: Reviewer two, point 9 - I also wondered about your answer to this point as I'm not sure you have actually addressed the issue and also if the third experiment doesn't actually address the RCI's then why is it included in this paper? Reframing the paper may help here.*

A: We understand the concern of the Reviewer about the link between resilience metrics and the empirical part of the paper (Sect. 5). We believe that the modifications presented under the Answers 9, 11, 13 have strengthened this link. Furthermore, in the third version of the manuscript, we have put more emphasis on the fact that we consider qualitative research as valuable for preliminary studies. Hence, we consider that these techniques shouldn't be excluded from a resilience assessment, but should be employed to support monitoring of RCI. We have stressed out this point with additions and modifications in Sect. 3.4 (p. 6 l. 17-20), Sect. 4 (p. 8 l. 12-13) and in the Conclusions (p.19 l. 1-5).

15)

*R3: P2 Line 4: This is too early to introduce the RCI's if they are a novel measure, you should base them within the literature.*

A: Since RCI is a novel term, we rephrased this sentence of the Introduction (p. 2 l. 3-4):

> 'i) the variables that are available in the context of a flood resilience project (RainGain) and that can be adopted as indicators of the impact of communication on urban resilience to climate risks.'

16)

*R3: P2 Line 17: a direct quote should have a page number.*

A: We thank the Reviewer for pointing at this omission. We have added the page number at p.2, l. 18.

17)

*R3: P3 Line 31: 'must be considered.' Considered for what?*

A: We have replaced this imprecise sentence with the following one (at p. 4 l. 5-6):

> 'According to this method, multiple spatial and temporal interacting scales most be considered to apprehend, for instance, how resilience at the level of a neighborhood could affect the resilience of a city.'

18)

*R3: P3 Section 3.1: I would like more references and a greater criticism of this section.*

Following the Reviewer's comment, we have added new references; we then identify the strengths and weaknesses of RA communication indicators. For a global review of the RA method, we prefer to refer to the PhD thesis by Vicari (to be defended on the 5/11/18) and to keep the focus of the paper on communication indicators so that the fluency of the paper is not reduced. The new references and the updated extracts with a critique of communication indicators are at l. 2 and l. 8-12 of p. 4.

19)

*R3: P4 Section 3.3: The Hyogo Framework is out of date, you should be using the more recent Sendai Framework.*

A: We definitely agree with the Reviewer on this point. Indeed, the previous version of the manuscript refers to the scientific report "Review of Alternative Approaches to Assess Resilience to Extreme Weather" (Vicari et al., 2015) that has been released when the Sendai framework was just launched and corresponding progress indicators didn't exist yet. Following the Reviewer's suggestion, we have updated Sect. 3.3 of the V3 paper.

20)

*R3: P4 Line 29: Indicators of knowledge and communication examples are listed, but how are these measured? Qual or quant? Do they support your method?*

A: We have included this information in Sect. 3.3 (p. 5 l. 3-13) and in Sect. 3.4 (p.5 l. 15-21), as suggested by the Reviewer.

21)

*R3: P5 Line 24: what is the principle of subsidiarity?*

A: Following the comment of the Reviewer, we have added Footnote 1 (at p. 3) with the definition of 'subsidiarity'.

22)
*R3: P6 Lines 6-23: are these referenced or are they your own measures? Also how are they measured, how can you use them to measure impact? Are they qual or quant? Why was the comparison factor included, what value does it bring?*

A: We thank the Reviewer for bringing out that we didn't clearly state the following points:
- We have defined these indicators on the basis of our former review 'State of the Field of Social Impact Assessment under the Paris – R100 Projection' (Vicari et al., 2016);
- For a matter of space, we provide examples of how some of these variables can be measured through the experiments presented in Sect. 5;
- All these indicators are based on quantitative variables;
- Comparison allows to observe how communication impacts change over time, in different locations and from one segment of the population to the other.

We have made explicit these four points with additions and modifications to Sect. 3.4 (p.5 l. 17-20 and 22-24), Sect. 4 (p. 7 l. 10-21 and l. 33-34; p. 8 l. 7-13), Sect. 5.1 (p. 11 l. 7-14), Sect. 5.2 (p. 11 l. 16 - p. 12 l. 3; p. 15 l. 4-12).

23)

*R3: P8 Figure 1: I don't think this is appropriate format to display this data, it is fairly confusing. Also the title states that that target values were compared with attained values, but how were these selected?*

A: We have removed this figure since it is not essential for the paper argumentation and, according to the Reviewer, it weakens the clarity of the paper. In Sect. 5 (p. 8 l. 26-30) of the third version of the manuscript, we have specified that the target values have been discussed and agreed with the European Commission officers in charge of the funding programme.

24)

*R3: P9, Line 5 - reported data on audience size of printed media distribution is generally not very reliable - additionally you cannot tell if participants who received the paper actually saw your article or not, so the limitations of this need to be acknowledged.*

A: Following the Reviewer's comment, we have included a reference to the limits of this assessment technique (p. 10 l. 7-10).

25)

*R3: P9, Line 8: What is WebTv?*

A: A Web TV is a television channel broadcast via the Internet. We have replaced this term with the more generic one 'online video reports' (p.9 l. 12).

26)

*R3: P10, Figure 3: I don't understand why the numbers of newspapers disinvited is increasing so dramatically, surely this is not related to the project as newspaper readership is usually fairly consistent?*

A: Following the Reviewer's comment, we noticed that the figure, that displays a cumulative curve, is not clearly explained. We have included a reference to the limit of this assessment technique, as indicated in Answer 24. Furthermore, we have rephrased the caption (p. 11) and the main text (p. 10 l. 10-15).

27)

*R3: P11, Line 8: I would like some description of the stimulus material here, what was in the exhibition?*

A: As suggested by the Reviewer, we have added these details in the Sect. 5.2 (p. 12 l. 4-8).

28)

*R3: P11, Line 23: why were so many participants who had not seen the exhibition included in this data set if it was an attempt to assess the impact of the exhibition?*

A: We agree that a larger number of exhibition visitors would have been suitable. This situation has been due to the limited size of the sample and to the fact that the invitation to answer the questionnaire was sent to all the students and workers of the school. We mention the limits of this experiment in Sect 5.2, at p. 12 l. 12-14.

29)

*R3: P13, Line 16: You cannot rely on 'a plausible explanation', that is just an assumption.*

A: Following the Reviewer's concern about this term, we have replaced it with 'We could assume' (p.14 l. 19).

30)

*R3: P13, Line 20: where had people heard about the project?*

A: As highlighted by the Reviewer, additional information is needed on this point that has been included in the Footnote 3 (p. 12).

31)

*R3: P13, Line 21: This is also an assumption, without clarifying some more of the variables for the reader I cannot be sure of this.*

A: Following the concern of the Reviewer, we have rephrased the sentence as it follows (p. 15 l. 2-3):

> 'We can suppose that face–to–face communication can strongly reinforce transmission of highly technical information.'

32)

*R3: P14, Section 5.3: This seems to be measuring the success of an individual comms project, rather than contributing to the broader aims of the study.*

A: We understand the Reviewer's point, we have addressed this concern in the Answers 9 and 11.

33)

*P15 and 16, Table 2, Table 3: I would like to see some reasoning for how these surveys were designed, as there seems to be several design flaws and leading questions.*

A: Following the Reviewer's comment, we have included this information in Supplement 2.

34)

*P16 Line 13: I would like an acknowledgement of the limitations of using only quant here.*

A: As requested by the Reviewer, we have stressed out this point through the following additions to the 'Conclusions': p. 18 l. 6-8 and l. 18-21; p. 19 l. 1-4.

[revised manuscript text omitted]

The questionnaire included questions on the professional background of the respondents. These questions allowed to exclude six experts from the sample, in order to obtain a relative homogeneity in terms of background knowledge. As a result, the final sample consisted in 31 respondents (see Supplement 1 for demographic data). Other questions were aimed at identifying through which source of information the respondents learnt about the project. On the basis of these questions the sample has been divided in four subsets: 1) 13 visitors to the exhibition; 2) five visitors who also read the brochure distributed at the exhibition; 3) six respondents who received only informal information (from word of mouth[3]); 4) 12 participants who never heard about the project. In order to perform a comparative experiment, the first subset has been considered as the experimental group with 13 respondents, while the third and fourth subsets have been considered as the control group with 18 respondents. We have used the Fisher's Exact test to compare the answers by the experimental group with those of the control group.
* * *
[3] We use the term 'from word of mouth' to refer to information that was passed from person to person – working or studying at the school – by oral and informal communication.

[Figure]

[Figure]

[Figure]

**c) "Why is it important to measure precipitations at small scale?"**

WRONG ANSWER: "To obtain reliable long term forecasts, i.e. up to one month in advance."

**Figure 3: The answers to three of the questionnaire questions on the RainGain exhibition held in April – May 2014. 100% corresponds to the total number of respondents included in each subset: 31 respondents in the first row, 13 respondents in the second row, 5 respondents in the third row, and so on.**

Figure 3 (a) shows that the number of respondents who visited the exhibition and have ticked the correct option for the question 'What is the spatial scale of the weather data provided by the radar?' is 23% higher than in the control group. As it appears in Fig. 3 (b), the wrong responses to the question 'What are the advantages of X band weather radars compared to C band and S band radars?' are 20% less frequent among the exhibition visitors. According to the results presented in Fig. 3 (c), the number of respondents who visited the exhibition and have provided a wrong response to the question 'Why is it important to measure precipitations at small scale?' is 15% lower than in the control group. The discrepancy between the visitors' results and the control group results is between 15% and 23% and it provides an approximate indication of the impact of the exhibition in terms of knowledge dissemination.

An unexpected result concerns the responses of the respondents who read the brochure at the exhibition in Fig. 3 (a) and 3 (c). In Fig. 3 (a) the rate of correct responses of the respondents who read the brochure is lower (60%) than in the experimental group (73%). Figure 3 (c) shows that the rate of wrong answers by the respondents who read the brochures is surprisingly high (40%): it is close to the rate of wrong answers by the respondents who never heard about the project (42%). We could assume that the respondents, who picked the brochure, have spent little time reading the exhibition panels and that part of the brochure information was not enough didactic and suitable for the general public.

Figure 3 (c) highlights another interesting result: the lowest rate of wrong answers corresponds to the group of respondents who didn't attend the exhibition but heard about the project from word of mouth. We can suppose that face-to-face communication can strongly reinforce transmission of highly technical information.

To sum up, the answer rates, displayed in Fig. 3, show that the exhibition had a modest positive effect on the respondents' awareness about a flood resilience project, the background environmental issues and the solutions being developed. We suppose that this effect was reinforced by word of mouth communication, but has also been weakened by the brochure.

Figure 4 presents the answers to a questionnaire question aimed at evaluating the risk perception and project acceptance of the respondents who visited the exhibition. The results show that the exhibition and the brochure, i.e. formal and official information, helped to reassure the respondents on security issues and encouraged them to support the implementation of the flood resilience project. Word of mouth communication didn't have such a positive effect as formal information, but neither did it compromise the achievement of the project goals.

The Fisher Exact test[4] has been applied to the results of the four questions: $p-values$ aren't significant, as these are always greater than 0.05 (the conventionally accepted significance level). Hence, the test confirms that, because of the small size of the sample, the differences between the answers by the experimental group and of the control group aren't statistically significant.
* * *
[4] We have computed a 2x2 contingency table, for each questionnaire question, with the frequencies of: a) the correct answers by the experimental group; b) the correct answers by the control group; c) the wrong answers by the experimental group; d) the wrong answers by the control group. We have then applied the Fisher's Exact test because in all the 2x2 contingency tables at least one value is $N \le 5$. The test uses the following formula where the 'a,' 'b,' 'c' and 'd' are the individual frequencies of the 2X2 contingency table, and 'N' is the total frequency:

$p = ((a + b)!(c + d)!(a + c)!(b + d)!)/a!b!c!d!N!$

rosa 17/10/18 19:36

rosa 15/10/18 14:16

rosa 15/10/18 14:19

rosa 15/10/18 14:16

rosa 17/10/18 19:36

rosa 15/10/18 14:48

rosa 23/10/18 13:58

rosa 23/10/18 13:59

rosa 23/10/18 14:00

rosa 23/10/18 13:59

rosa 17/10/18 19:34

rosa 17/10/18 19:35

rosa 17/10/18 19:34

rosa 17/10/18 19:37

rosa 23/10/18 13:34

rosa 17/10/18 19:34

rosa 15/10/18 14:45

rosa 23/10/18 13:29

rosa 23/10/18 13:28

rosa 23/10/18 13:29

rosa 23/10/18 13:29

[revised manuscript text omitted]

rosa 17/10/18 20:41

rosa 17/10/18 19:35

rosa 18/10/18 13:42

rosa 17/10/18 19:35

rosa 17/10/18 20:49

rosa 17/10/18 20:58

rosa 17/10/18 20:58

rosa 18/10/18 14:28

---

## Author Response (AR3)

Many thanks to the Editor for guiding the review and to Reviewer 3 for the time devoted to the review and his final helpful comments. We have addressed all the technical issues raised by the Reviewer and we believe that we have met his requirements.

ANSWER TO REVIEWER 3

*Reviewer 3: Thank you to the authors for responding to my suggestions in such a comprehensive way. The paper is now much clearer to me and I am much better able to follow the statements of design and context. I was also very pleased to see clarification of the method, with clear reference to the limitations encountered during the process, which gave me a much better insight into the study.*

Authors: The authors would like to thank the Reviewer for his feedback and we are glad to read that he appreciated the improvements that we have introduced in the last version of the manuscript and that we have met his expectations.

1)

*R3: I appreciate the addition of references to existing research in risk communication on page 5 from line 28 onwards, but I would like more than just a list of names here - even a summary sentence would be suitable, such as: 'The first series of works explore the impact of risk communications and effectiveness (ref ref erf) who found that....' and 'Second set... methodologies (ref ref ref) which included...'.*

A: We agree with the Reviewer that further details, on the research work we refer to, are needed. We have included complementary information at p. 6 (l. 3-13)

2)

*R3: Some additional small technical corrections:*
*Pg 1 line 17 and line 21: I think perhaps you mean 'comprehend' not 'apprehend'?*

A: We thank the Reviewer for pointing at this inaccurate use of the verb 'apprehend', we have replaced it with 'comprehend' at p. 1 (l. 12,17,21), p. 4 (l. 8), p. 6 (l. 15).

3)

*R3: Pg 2 line 3: NWE IVB this is the first use of this acronym*

A: Following the comment of the Reviewer, we have added Footnote 1 (at p. 2) that explains this acronym:

> INTERREG IVB North West Europe "is a financial instrument of the European Union's Cohesion Policy. It funds projects which support transnational cooperation" (Interreg NWE IVB: http://4b.nweurope.eu/, last access: 20 November 2018).

4)

*R3: Pg 3 footnote 1: Is the reference for subsidiarity (thank you for explanation) also Tanguy 2013?*

A: Following the Reviewer's comment, we have added the following reference: Oxford English Dictionary: https://en.oxforddictionaries.com/, last access: 20 November 2018. Indeed, our explanation of subsidiarity is based on the definition by the Oxford English Dictionary.

5)

*R3: Pg 4 line 1: check reference style for a website*

A: We thank the Reviewer for pointing at this reference style inaccuracy. We have corrected all the website references at p. 2 (footnote 1), p. 3 (footnote 2), p. 4 (l. 4), p. 5 (l. 3, footnote 3), p. 6 (l. 27), p. 9 (l. 19).

6)

*Pg 6 line 6 and line 13: Reference the year for Neresini and Pellegrini*

A: Following the comment of the Reviewer, we have added the year after both references to Neresini and Pellegrini.

7)

*Pg 8 line 33: I think this should be advice not advise*

A: We thank the Reviewer for pointing at this mistake that has been corrected.

[revised manuscript text omitted]